# Trioxane-based MS-cleavable cross-linking mass spectrometry for profiling multimeric interactions of cellular networks

Clinton Yu[1], Eric Novitsky[2], Sree Ganesh Balasubramani[3], Xiaorong Wang[1], Xiyu Shen [1], Qin Yang[1,4], Scott Rychnovsky [2], Ignacia Echeverria [3,5] & Lan Huang [1] ✉

Cross-linking mass spectrometry (XL-MS) is a powerful technology for mapping protein-protein interactions (PPIs) at the systems level. While bivalent cross-links are effective for defining protein interactions and structures, multivalent cross-links offer enhanced spatial resolution to facilitate characterization of heterogeneous protein complexes. However, their identification remains challenging due to fragmentation complexity and the vast expansion of database search space. Here, we present tris-succinimidyl trioxane (TSTO), a novel trioxane-based, MS-cleavable homotrifunctional cross-linker capable of targeting three proximal lysines simultaneously. TSTO's unique MS-cleavability enables concurrent release of cross-linked peptide constituents during collision-induced dissociation, permitting their unambiguous identification. The TSTO-based XL-MS platform is effective for mapping cellular networks from intact cells and tissues, illustrating its versatility for complex biological systems. Trimeric interactions captured by TSTO reveal structural details inaccessible to bifunctional reagents, enhancing modeling accuracy and precision. Furthermore, this development opens a new avenue for designing multifunctional MS-cleavable cross-linkers to further advance structural systems biology.

Cross-linking mass spectrometry (XL-MS) has emerged as a transformative technology for interactomics and structural proteomics[1–4]. By covalently linking proximal amino acid residues within or between proteins, XL-MS provides unique insights into the architecture of protein complexes and the spatial arrangement of individual protein domains. Due to various innovations, this methodology has revolutionized structural biology by complementing traditional structural techniques and offering a more holistic view of protein assemblies in their native environments[4–8]. Over the years, XL-MS has significantly evolved with the incorporation of various functionalities and targeting groups in cross-linker design, broadening its applications and enhancing the scope of XL-MS studies. In particular, diverse MS-cleavable cross-linking reagents have been developed to facilitate the detection and identification of cross-linked peptides[1,3,4], enabling proteome-wide analyses of cellular networks to elucidate their structural organization from intact cells, subcellular organelles, tissues and clinical samples.

Up to now, the majority of cross-linking reagents are bifunctional (carrying two reactive groups), and current XL-MS analyses have focused on the identification of cross-links between two residues to infer pair-wise interactions. While bivalent cross-links are effective for

[1]Department of Physiology & Biophysics, University of California, Irvine, Irvine, CA, USA. [2]Department of Chemistry, University of California, Irvine, Irvine, CA, USA. [3]Department of Bioengineering and Therapeutic Sciences, University of California, San Francisco, San Francisco, CA, USA. [4]Department of Medicine, University of California, Irvine, Irvine, CA, USA. [5]Quantitative Biosciences Institute, University of California, San Francisco, San Francisco, CA, USA. ✉e-mail: lanhuang@uci.edu

PPI mapping, multivalent cross-links can offer enhanced spatial resolution and provide additional distance restraints to facilitate structural analysis of protein complexes. Furthermore, these higher-order cross-links can uncover interaction sites and conformations that are not easily detectable with binary cross-linkers. The resulting multimeric interactions can lead to the discovery of new functional insights and regulatory mechanisms within protein complexes, especially for those dynamic and heterogeneous ones. Although multivalent cross-links can be formed by bifunctional cross-linkers, the difficulty of their identification has rendered them largely invisible. Due to the additional expansion of the search space associated with each successive peptide, traditional non-cleavable cross-linking reagents are incapable of this task. As a result, MS-cleavable cross-linking reagents are critical for alleviating this issue by physically separating cross-linked constituents to facilitate their individual identifications[1,9,10]. During the course of our study, a homotetrafunctional MS-cleavable cross-linking reagent utilizing four NHS ester-targeting groups (aka Bisby) was reported[11]. However, the additional bonds required for cross-linking each successive peptide complicate cross-link fragmentation and identification. Therefore, new approaches are needed to permit effective identification of multivalent cross-links, especially for proteome-wide PPI profiling.

To this end, we aimed to design an MS-cleavable trifunctional cross-linker capable of simultaneously targeting three residues, while more importantly streamlining peptide identification through a core structure that enables synchronous release of all cross-linked peptides in a single step. This will allow us to expand the detection of protein connectivity and provide additional restraint information for improved structural elucidation at the systems level. Here, we present the design, synthesis and characterization of a novel membrane-permeable, MS-cleavable homotrifunctional cross-linker, TSTO (tris-succinimidyl trioxane), to enable simultaneous capture and identification of trimeric PPIs. XL-MS analysis of human 26S proteasomes has demonstrated that TSTO is effective in cross-linking protein complexes and that the resulting cross-linked peptides display unique and predicable fragmentation during collision-induced dissociation (CID), enabling their simplified and accurate identification using multistage mass spectrometry (MS$^n$). Importantly, we have captured trimeric interactions to better define interfaces between proteasome subcomplexes. Moreover, TSTO has been successfully applied for in vivo cross-linking of HEK293 cells and mouse heart tissues, demonstrating its applicability in elucidating cellular networks. Apart from binary interactions, trimeric interactions captured by TSTO cross-linking facilitated the characterization of protein oligomers and enhanced structural modeling with improved accuracy and precision. In summary, we show that TSTO can uncover multimeric interactions to yield more detailed PPI networks, advancing our understanding of cellular processes and biological function.

## Results

### Developing a Trioxane-based MS-cleavable Homotrifunctional Cross-linker TSTO

In order to capture trimeric interactions, we designed a trioxane-based cross-linker TSTO and accomplished its synthesis through five steps (Fig. 1a). As shown, TSTO carries a unique symmetrical structure comprising three NHS esters connected via a central trioxane group[12,13] (Fig. 1a), permitting concurrent cross-linking between three lysine residues to form a trivalent cross-link among three individual peptides (aka tripeptide tri-link, [α,β,γ]) (Fig. 1b, Type I). In comparison to a traditional cross-link between two individual peptides, accurate identification of a tri-link would be much more challenging due to further expansion of the database search space ($n^3$). Therefore, the design of the central trioxane is crucial as it carries three equal MS-cleavable bonds that are weaker than peptide bonds and can be cleaved in parallel using CID to simultaneously release all three cross-linker arms

in a single step. This leads to physical separation of the three cross-linked peptide constituents, yielding three fragment ions during MS$^2$ analysis that can be subjected to MS$^3$ sequencing (Fig. 1b, Type I). As shown, trioxane cleavage results in an identical and defined aldehyde remnant (AR) on each peptide constituent, allowing for their unambiguous identification. In addition, this minimizes the total number of MS$^2$ fragments, simplifying ion selection for subsequent MS$^3$ analysis. In addition to tripeptide TSTO cross-links, tri-links can be formed between two peptides, linking one lysine in one peptide (α) and two lysines in another peptide (β) (aka dipeptide tri-link, [α-β$_2$]) (Fig. 1b, Type II). For dipeptide tri-links, two fragment ions would be observed in MS$^2$, one corresponding to a peptide carrying a single AR-modified lysine and the other representing a peptide carrying two AR-modified lysines. Finally, TSTO cross-linking can yield traditional cross-links in which two lysines from two different peptides are cross-linked while the third NHS ester of TSTO is hydrolyzed (aka, dipeptide bi-link, [α-β]) (Fig. 1b, Type III). MS$^2$ analysis of a dipeptide bi-link would yield two fragment ions corresponding to peptides each carrying a single AR-modified lysine, while the hydrolyzed arm is released as a neutral loss. In addition to TSTO inter-links formed by two or three peptides as described above, monomeric cross-linked species containing a single peptide including 'intra-links' and 'dead-ends' can occur due to the hydrolysis of one or two NHS esters of TSTO. Notably, TSTO inter-linked peptides are the most structurally informative products for PPI mapping, and thus the focus of our analysis. Owing to their unique MS-cleavability, TSTO cross-linked peptides can be identified using the same LC-MS$^n$ workflow that has been previously established for our sulfoxide-containing MS-cleavable cross-linkers[1,4,14].

### Characterization of TSTO Cross-linked Synthetic Peptide by MS$^n$ Analysis

We first characterized TSTO cross-linking on the synthetic peptide Ac-SR8 (Ac-SAKAYEHR). Under our experimental conditions, three Ac-SR8 cross-linked products were detected: dead-end modified Ac-SR8 (α$_{DN}$), inter-linked Ac-SR8 homodimer [α-α], and Ac-SR8 homotrimer [α, α, α]. MS$^2$ analysis of dead-end modified Ac-SR8 (m/z 667.3163$^{2+}$) yielded two dominant ions (m/z 542.2656$^{2+}$, 551.2708$^{2+}$) (Supplementary Fig. 1a). MS$^3$ peptide fragment analysis determined these ions to be AR-modified Ac-SR8, with the lower mass ion corresponding to an AR moiety undergoing water loss (namely AR*), resulting in the detection of an ion doublet (α$_{AR}$ and α$_{AR*}$) with mass difference (Δ) of 18.02 Da (Supplementary Fig. 1b, c). Two differently-charged species of inter-linked Ac-SR8 homodimer [α−α] were detected (m/z 773.7071$^{3+}$, 580.5319$^{4+}$), each fragmenting into dominant ions corresponding to α$_{AR}$ and α$_{AR*}$ during MS$^2$ analysis (Supplementary Fig. 2a, b). Similarly, MS$^2$ analysis of the Ac-SR8 homotrimer tri-link [α, α, α] (m/z 826.6508$^{4+}$) yielded three dominant ions (α$_{AR}^{2+}$/α$_{AR}^{2+}$/α$_{AR}^{1+}$) (Supplementary Fig. 2c). While MS$^3$ analyses of both α$_{AR}$ and α$_{AR*}$ resulted in their unambiguous identification (Supplementary Fig. 1b, c), selecting AR*-modified peptides for sequencing would be preferred due to the AR moiety's propensity for dehydration.

### Characterization of TSTO Cross-linked BSA by MS$^n$ Analysis

To characterize TSTO cross-linking in proteins, we performed XL-MS analysis on the model protein BSA, focusing on TSTO inter-linked peptides. As a result, all three types (I-III) of TSTO inter-linked peptides were identified by LC MS$^n$, each displaying the characteristic MS$^2$ fragmentation as expected. This is illustrated by MS$^n$ analyses of representative TSTO cross-linked BSA peptides (Supplementary Fig. 3). For a tripeptide tri-link [α, β, γ] (m/z 795.7120$^{6+}$), its MS$^2$ analysis yielded three sets of dominant ions corresponding to α$_{AR}$/α$_{AR*}$, β$_{AR}$/β$_{AR*}$, and γ$_{AR}$/γ$_{AR*}$ fragments (Supplementary Fig. 3a). As shown, MS$^3$ analyses of the three cross-link fragments α$_{AR}$ (m/z 638.3130$^{2+}$), β$_{AR*}$ (m/z 773.8743$^+$), and γ$_{AR*}$ (m/z 947.9347$^+$) identified a tripeptide TSTO tri-link among BSA lysines K228, K374, and K498

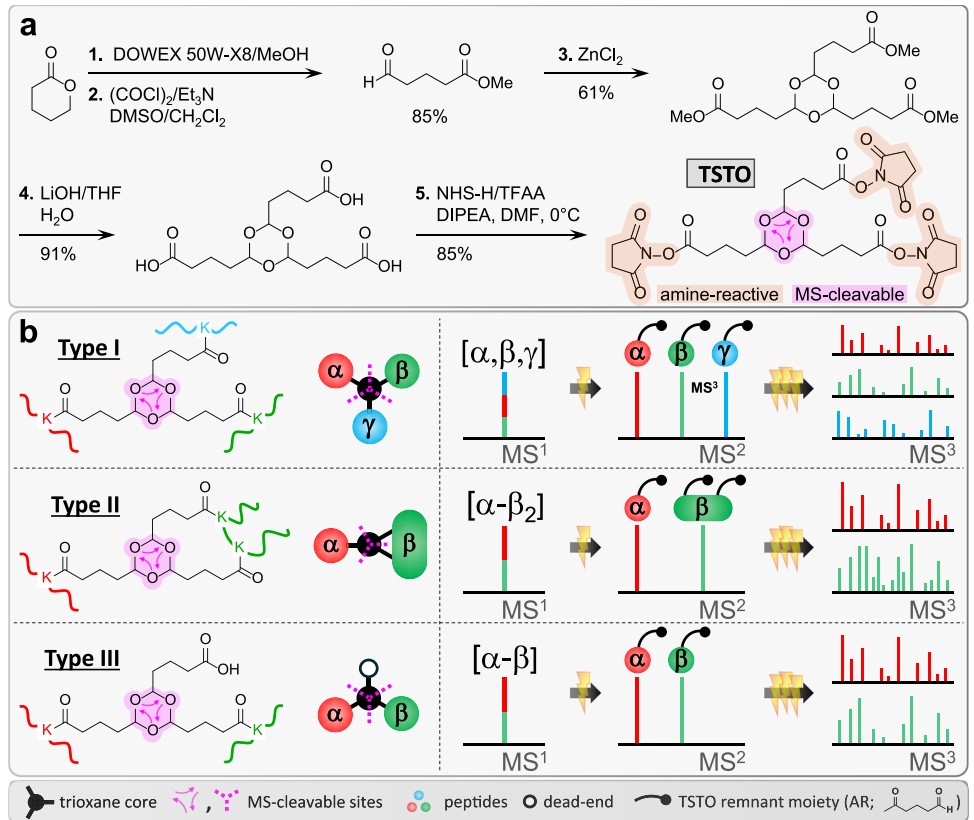

**Fig. 1 | TSTO design, synthesis, and expected cross-link formation and MS^n fragmentation. a** TSTO synthesis. **b** TSTO cross-link products and their expected fragmentation during CID analysis. For tripeptide tri-links (Type I), MS² yields three fragment peptides with single AR-modified lysines. Dipeptide tri-links (Type II) produce two fragment peptides, one with a single AR-modified lysine and one with two. Finally, dipeptide bi-links (Type III) result in two fragment peptides, each with a single AR-modified lysine. MS³ analyses of these fragments enable unambiguous identification of TSTO tri- and bi-links.

(Supplementary Fig. 3b). MS² analysis of a dipeptide tri-link [α-β₂] (m/z 1054.7530⁴⁺) resulted in two sets of dominant ion species: α_AR/α_AR*, and β₂AR/β_AR_AR*/β₂AR* (Supplementary Fig. 3c). The detection of a fragment triplet with 18 Da increments indicates that peptide β carries two modified lysines, whereas peptide α only contains a single modified lysine. MS³ analyses of α_AR* and β₂AR* identified their sequences as ³⁷²LAK_AR*EYEATLEECCAK³⁸⁶ and ⁴⁹⁰TPVSEK_AR*VTK_AR*CCTESLVNR⁵⁰⁷, respectively (Supplementary Fig. 2d), signifying a dipeptide tri-link [BSA:K374 - BSA:K495, K498]. Finally, for a dipeptide bi-link [α-β] (m/z 992.7017⁴⁺), MS² fragmentation produced two dominant ion pairs: α_AR/α_AR*, and β_AR/β_AR* (Supplementary Fig. 3e); MS³ analyses of α_AR* and β_AR* determined a cross-link between BSA:K117 and BSA:K489 (Supplementary Fig. 3f).

In total, 823 redundant cross-link spectra matches (CSMs) were identified, corresponding to 167 unique ones (Supplementary Data 1). Of these, 21 were tripeptide tri-links, 24 were dipeptide tri-links, and 122 were dipeptide bi-links. Overall, tri-links contributed ~ 27% (45/167) of the total unique CSMs. Breaking down tripeptide and dipeptide tri-links into their respective constituent residue pairs, a combined total of 118 K-K pairs were identified, with 37 being contributed by both TSTO tri- and bi-links, whereas 50 and 31 were unique contributed by TSTO bi-links and tri-links, respectively.

Considering the spacer arm length of TSTO (~ 14 Å), lysine residues with Cα-Cα distance ≤ 35 Å were expected to be preferentially cross-linked. When mapped to the high-resolution crystal structure of BSA (PDB:4F5S) (Supplementary Fig. 4a), the overall mapped distance median was 21.1 Å with a satisfaction rate of cross-links under ≤ 35 Å of 90%. Taken together, these results demonstrate that TSTO is effective for protein cross-linking, and the resulting cross-linked peptides exhibit unique MS² fragmentation patterns

that are both predictable and reliable for unambiguous identification by LC MS^n analysis.

## TSTO XL-MS Analysis of the 26S proteasome

To explore TSTO's capability in XL-MS analysis of protein complexes, we performed TSTO cross-linking of affinity-purified human 26S proteasomes. Similar to BSA, all three types of TSTO inter-linked peptides of the 26S proteasome were detected (Fig. 2). This was exemplified by MS^n analyses of a representative tripeptide tri-link [α, β, γ] (m/z 998.9354⁵⁺) among Rpt1:K116, Rpt4:K206, and Rpt3:K238 (Fig. 2a, b), a dipeptide tri-link [α-β₂] (m/z 1209.0961⁴⁺) [Rpn12:K281-Rpn3:K455, K461] (Fig. 2c, d), and a dipeptide bi-link [α-β] (m/z 773.4180⁴⁺) between Rpt4:K72 and Rpt6:K222 (Fig. 2e, f). Cumulatively, TSTO-based XL-MS analysis of the 26S proteasome resulted in a total of 808 unique CSMs (Supplementary Data 2). Of these, 41.3% were tri-links, indicating that TSTO can efficiently cross-link three spatially proximal residues within the proteasome. To assess the accuracy of TSTO cross-links, we mapped them to a high-resolution 26S proteasome structure (PDB: 7QY7[15]). As a result, all of the mapped tri-links and bi-links have ~ 90% satisfaction rates (≤35 Å), supporting their validity (Supplementary Fig. 4). To understand whether the introduction of a supplementary reactive group is beneficial for describing protein topologies, we constructed a TSTO XL-PPI map of the 26S proteasome comprising 92 edges (Fig. 3a). Compared to previous XL-MS studies using bifunctional linkers[16,17], 31 additional inter-subunit PPIs were identified, with most of them at interaction interfaces between proteasomal subcomplexes. To better understand the benefits of TSTO over conventional reagents, we next examined the interactions captured by tri-links. Among the 35 trimeric interactions within the proteasome, 13 described multimeric interactions among the six ATPase subunits

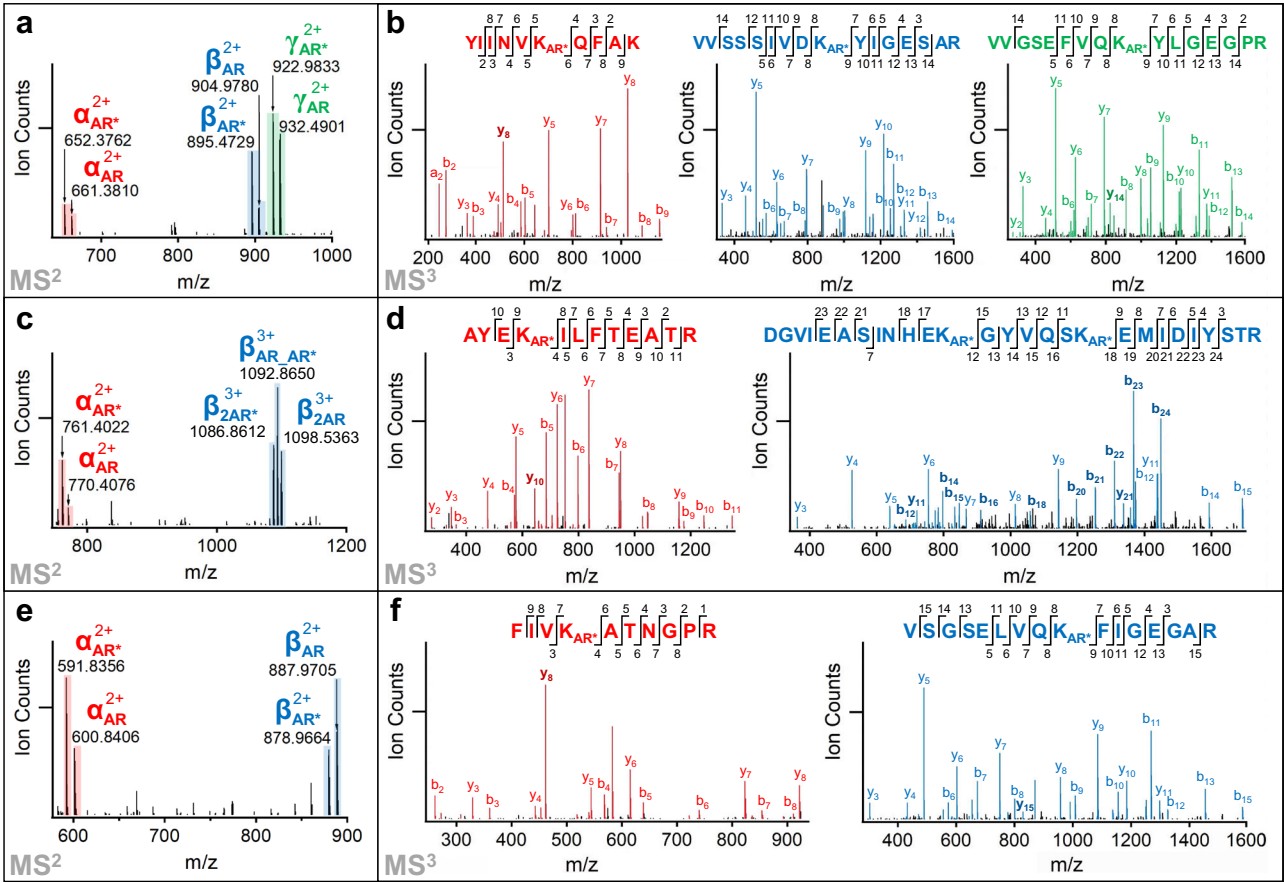

**Fig. 2 | Representative MS[n] analyses of the three types of TSTO cross-links identified from 26S proteasomes. a** MS[2] fragmentation of a tripeptide tri-link [α, β, γ] (*m/z* 998.9354[5+]) yielded a series of dominant ion doublets corresponding to $\alpha_{AR}/\alpha_{AR*}$, $\beta_{AR}/\beta_{AR*}$, and $\gamma_{AR}/\gamma_{AR*}$ peptides. **b** MS[3] analyses of the cross-link fragments $\alpha_{AR*}$ (*m/z* 652.3762[2+]), $\beta_{AR*}$ (*m/z* 895.4729[2+]), and $\gamma_{AR*}$ (*m/z* 922.9833[2+]) identified them as [110]YIINVK$_{AR*}$·QFAK[120], [198]VVSSSIVDK$_{AR*}$·YIGESAR[213], and [230]VVGSEFVQK$_{AR*}$·YLGEGPR[245], signifying a tripeptide tri-link among Rpt1:K116, Rpt4:K206, and Rpt3:K238. **c** MS[2] fragmentation of a dipeptide tri-link [α-β₂] (*m/z* 1209.0961[4+]) yielded two dominant sets of ions—a doublet representing $\alpha_{AR}/\alpha_{AR*}$ and a triplet representing $\beta_{2AR}/\beta_{AR\_AR*}/\beta_{2AR*}$ peptides. **d** MS[3] analyses of the cross-link fragments $\alpha_{AR*}$ (*m/z* 761.4022[2+]) and $\beta_{2AR*}$ (*m/z* 1086.8612[2+]) identified their sequences as [278]AYEK$_{AR*}$·ILFTEATR[289] and [444]DGVIEASINHEK$_{AR*}$·GYVQSK$_{AR}$·EMIDIYSTR[470] respectively, signifying a dipeptide tri-link among Rpn12:K281, Rpn3:K455, and Rpn3:K461. **e** MS[2] fragmentation of a dipeptide bi-link [α-β] (*m/z* 773.4180[4+]) resulted in two dominant ion doublets $\alpha_{AR}/\alpha_{AR*}$ and $\beta_{AR}/\beta_{AR*}$. **f** MS[3] analyses of the cross-link fragments $\alpha_{AR*}$ (*m/z* 591.8356[2+]) and $\beta_{AR*}$ (*m/z* 878.9664[2+]) identified their sequences as [69]FIVK$_{AR*}$ATNGPR[78] and [214]VSGSELVQK$_{AR*}$FIGEGAR[229], signifying a pair-wise cross-link between Rpt4:K72 and Rpt6:K222. Note: AR: aldehyde remnant moiety; AR*: aldehyde remnant moiety after water loss (i.e., AR-H₂O). Source data are provided as a Source Data file.

(Rpt1-6), illustrating their proximity within the 19S base subcomplex (Fig. 3b). In addition, the connectivity between the 19S and 20S subcomplexes was described by three trimeric interactions, including one among Rpt4, α1 and α7, and one between Rpt1 and α4 (Fig. 3c, d). Apart from trimeric interactions involving three proteins, TSTO was able to concurrently place two distant residues of one protein (α4: K27, K166) to another (Rpt1:K418), due to their proximity in the three-dimensional structure. Another tri-link involving Rpt6 and α3 exemplifies the capability of TSTO to capture interactions involving adjacent residues (Rpt6:K397, K402) that may be missed by bifunctional linkers due to the tendency to form intra-links (loop-links). Moreover, TSTO cross-links placed a small proteasome subunit Dss1 in proximity to Rpn3, Rpn6, and Rpn7 within the 19S lid (Fig. 3e and Supplementary Fig. 6), which has not been reported by previous XL-MS studies of the human 26S proteasome[16,17]. Nonetheless, these interactions were supported by the two human 26S structures (PDB: 6MSB[18] and 7QY7[15]) as all cross-links between Dss1 and Rpn3, Rpn6, and Rpn7 have Cα-Cα distances ≤26 Å, falling below the distance threshold (Fig. 3E and Supplementary Fig. 5). Collectively, these results have demonstrated that TSTO is effective in capturing multimeric interactions of proteasome complexes, providing three-dimensional contacts for the first time to support the spatial organization of these proximal subunits.

## Integrative structure modeling using synthetic and experimental XL-MS data

To determine the value of trivalent versus bivalent cross-links, we applied integrative structure modeling to compute the structure of the proteasome base subcomplex (subunits Rpt1–Rpt6 and Rpn2). First, we generated synthetic datasets for the proteasome base subcomplex, replicating the experimental distance distribution of the proteasome TSTO dataset (Supplementary Data 2) to systematically compare trifunctional and bifunctional cross-linkers. Subsets of 20, 30, 40, 60, and 120 cross-links were randomly sampled based on the distance probability distribution, with at least six replicates created for each subset size. For this analysis, a cross-linking site was defined as the set of residues bridged by a single cross-linker. To illustrate the uniqueness of trifunctional linkages, we have focused our analysis on the comparison of trivalent and bivalent cross-links for integrative modeling. For each trivalent cross-link, a corresponding bivalent cross-link utilizing two of the three cross-linked lysines was used to ensure that the synthetic data reflects the spatial restraints of each cross-linker, providing a robust foundation for structural modeling.

Ensembles of the proteasome subcomplex configurations that satisfy the input information (i.e., the model) were found by exhaustive Monte Carlo sampling guided by the scoring function, starting with

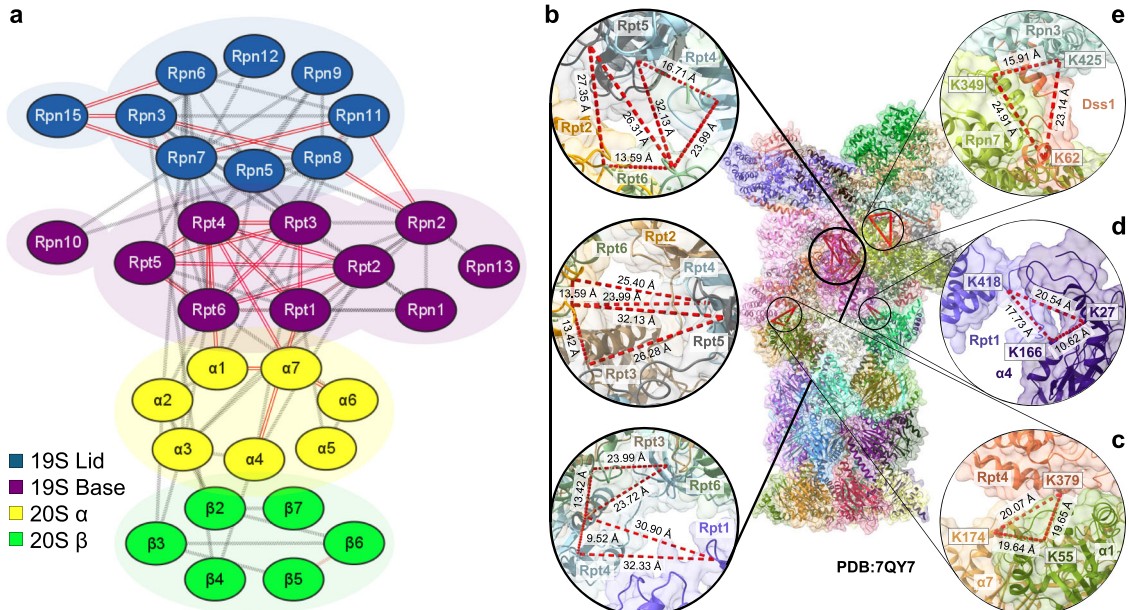

**Fig. 3 | TSTO XL-MS analysis of human 26S proteasome. a** XL-PPI map of 26S cross-links. In total, 32 subunits were identified participating in inter-subunit interactions—19 from the 19S subcomplexes and 13 from the 20S subcomplex. 19S lid subunits shown in blue, 19S base subunits in purple, 20S α ring subunits in yellow, and 20S β ring subunits shown in green. Red lines represent interactions that have been captured as tri-links; gray lines represent other cross-links. Mapping of trimeric interactions to a high-resolution 26S proteasome structure: (**b**) lining the inner pore between Rpt subunits of the AAA-ATPase ring (19S base subcomplex), (**c**, **d**) Interactions describing connectivity of 19S base subcomplex subunits to the 20S, and (**e**) Dss1 with 19S lid subunits Rpn3 and Rpn7.

random initial configurations of the rigid components. The models computed using trivalent cross-links were generally more accurate and precise than the ensembles computed using bivalent ones. The accuracy is defined as the average Cα root-mean-square deviation (RMSD) between the cryo-EM structure (PDB ID: 5GJR) and each of the structures in the ensemble, while the precision is defined as the average RMSD between all solutions in the ensemble. Increasing the number of both trivalent and bivalent cross-links used for modeling improved model accuracy and precision, with diminishing returns observed as the number of cross-links increased from 40 to 120. For datasets containing bivalent cross-links, the average model accuracy plateaued at ~ 9.8 Å, while for trivalent cross-links, the plateau occurred at ~ 8.7 Å. The trivalent cross-linking data consistently produced better model accuracies, with average RMSDs of 19.7, 14.1, 9.7, 8.8, and 8.7 Å for datasets with 20, 30, 40, 60, and 120 cross-links, respectively. In contrast, the bivalent cross-linking data resulted in higher average RMSDs of 25.3, 19.0, 13.3, 9.7, and 9.8 Å for the same subset sizes. A similar trend was observed for the cluster precisions (Fig. 4). Furthermore, we computed the structure of the base subcomplex of the 26S human proteasome using integrative modeling with experimentally derived DSSO and TSTO cross-linking datasets. The DSSO dataset included 83 cross-links within the base subcomplex[17]. The TSTO dataset comprised 18 trivalent and 143 bivalent cross-links; from the TSTO data, 65 bivalent cross-links were randomly selected (five replicates) to ensure that the total number of cross-linking sites was consistent between the DSSO and TSTO datasets. Models computed using the TSTO dataset were considerably more accurate than models computed with the DSSO dataset (15.3 Å vs 33.2 Å). These results highlight TSTO's ability to provide richer structural information, such as the spatial positioning of three protein regions, which is critical for generating accurate structural models.

### In Vivo TSTO Cross-linking of the HEK293 Cells

We next investigated whether TSTO was applicable for system-wide XL-MS analysis to delineate cellular networks. To this end, we performed TSTO cross-linking of HEK293 cells stably expressing HBTH-tagged CSN2, a subunit of the COP9 signalosome. To evaluate TSTO in-cell cross-linking efficiency, immunoblot analysis was carried out to probe the formation of CSN2-containing cross-linked products, which were represented by high molecular weight protein bands. Based on the formation of CSN2-containing oligomer and its increased abundance with increasing cross-linker concentration, TSTO was determined to be membrane-permeable and suited for in-cell cross-linking (Supplementary Fig. 7). The general TSTO-based in vivo XL-MS workflow is illustrated in Fig. 5, in which cross-linked peptides were subjected to two-dimensional peptide separation prior to LC-MS$^n$ analysis[19]. Across two biological replicates, we identified a total of 9079 unique CSMs (Supplementary Data 3), of which 32.3% were tri-links. As shown, TSTO tri-links increased the total PPI yield by ~ 27% compared to bi-link data alone (Supplementary Fig. 8a). Altogether, TSTO in-cell cross-linking yielded an XL-proteome of 1512 proteins containing 1242 PPIs (Supplementary Fig. 8b). These results indicate that TSTO cross-linking of intact cells is effective, and the presence of tri-links remains abundant in increasingly complex systems. Importantly, accurate identification of TSTO cross-links at the systems level was achieved using LC-MS$^n$.

To examine the validity of the TSTO cross-links, we first mapped them to available high-resolution structures of protein complexes identified here (Supplementary Fig. 8c)[20]. In total, 1790 K-K linkages were mapped across 539 CORUM complexes and 95% of them were considered satisfactory (≤35 Å). Next, we performed gene ontology (GO) analysis and confirmed that the TSTO XL-proteome covers a wide range of molecular functions, biological processes, and cellular components (Supplementary Fig. 9a). Compared to BioGRID and BioPlex databases, 48% of the TSTO XL-PPIs were known and 52% were novel. The STRING scores were found for 501 of the XL-PPIs, and ~ 70% were determined to be above 0.8 (Supplementary Fig. 9b), indicating high-confidence interactions[19,20]. Overall, among the 1242 inter-protein PPIs identified within this TSTO in vivo dataset, roughly one-third were novel when compared to an aggregate of recent systems-level cross-linking studies[19,21–24]. This suggests that while most XL-PPIs are supported, TSTO provides information that was not detected by

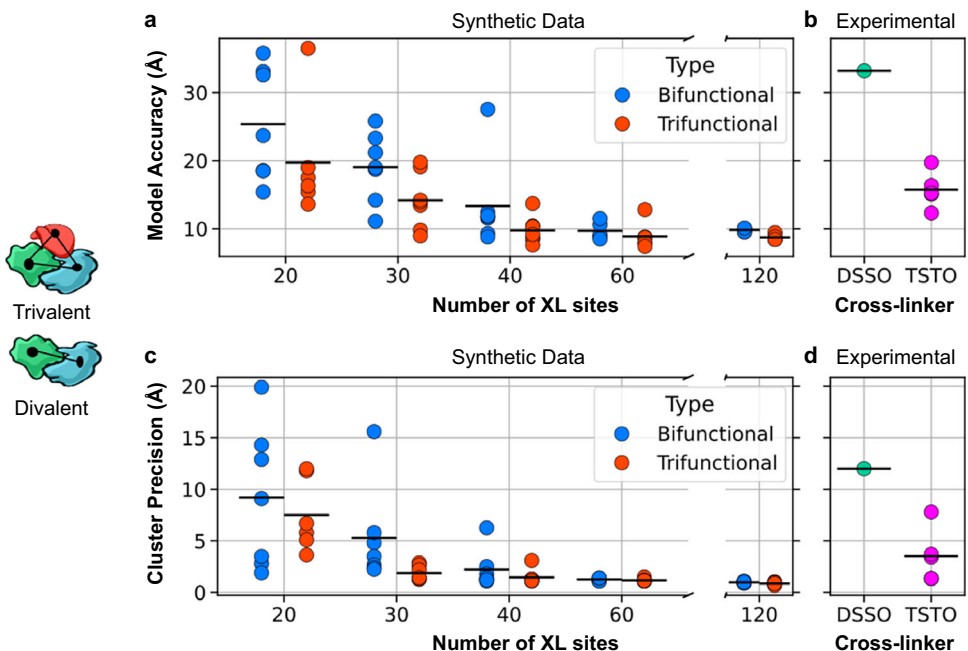

**Fig. 4 | Integrative structural modeling of the base subcomplex of the human 26S proteasome using synthetic and experimental XL-MS data. a** Model accuracy and (**b**) cluster precision were assessed as a function of the number of cross-links, using six replicates per condition with synthetic bifunctional and trifunctional cross-linker data. Black lines indicate mean values for each subset. **c** Model accuracy and (**d**) cluster precision were compared for models generated using TSTO (18 trivalent + 65 bivalent) and DSSO (83 bivalent) cross-linking data. For experimental TSTO data in (**b**) and (**d**), five datasets were generated, each combining all 18 trivalent with 65 randomly selected (from 143 available) bivalent cross-links. Source data are provided as a Source Data file.

bifunctional MS-cleavable reagents. Finally, to estimate the dynamic range of the XL proteome captured by TSTO cross-linking, we plotted the abundance distribution of the identified cross-linked proteins based on their copy numbers as determined by shotgun proteomics[25] (Supplementary Fig. 10). Compared to the MS-proteome, the TSTO XL-proteome was shifted towards higher abundance proteins, however, it is comparable to previous proteome-wide XL-MS studies[19,20,26]. Nevertheless, TSTO is capable of targeting cellular proteins across all cellular compartments and capturing interactions among proteins spanning five orders of magnitude. Collectively, these results demonstrate that TSTO is effective for global profiling of endogenous PPIs from their cellular environments.

## Multimeric Interactions of In Vivo Protein Complexes

Importantly, TSTO XL-MS has enabled the identification of multimeric interactions within various protein complexes. One well-represented complex is the 80S ribosome machinery. Specifically, extensive interactions describing subunit proximities were identified, particularly those between the 40S or 60S subcomplexes. This allows for an in-depth description of ribosomal PPIs with 3-D contacts (Supplementary Fig. 11). Interestingly, trimeric interactions were also found between 80S and several putative ribosome-binding partners, providing structural insights underlying their functional relevance in protein synthesis. Of these, the most frequent was SERBP1 (SERPINE mRNA-binding protein 1), which was shown to interact with 60S ribosome subunits RPL7A, RPL27, and RPL34 through several tri-links. While eukaryotic SERBP1 (Stm1 in yeast) has been associated with dormant ribosomes due to its role in clamping 40S and 60S subunits together to prevent mRNA access, it has been shown to primarily contact 40S subunits[27–29]. Interestingly, TSTO identified ribosome-SERBP1 cross-links spanning the 80S, including distant 60S subunits. In current high-resolution structures of ribosome-bound SERBP1, the majority of SERBP1 is unresolved, likely buried within the 80S. Together, the results suggest that the interaction of SERBP1 with ribosomal proteins is extensive, contacting various 40S and 60S subunits to inactivate 80S machinery.

Another trivalent cross-link was identified between 40S ribosome subunits and UBAP2L (Ubiquitin-associated protein 2-like). While UBAP2L is a known RNA-binding protein (RBP) that may associate with ribosomes to facilitate protein synthesis[30], its function and roles remain poorly characterized. TSTO identified a trimeric interaction between UBAP2L and neighboring 40S subunits RPS7 and RPS27, tri-angulating its position near the small ribosomal complex and correlating well with the role of the 40S in initial binding and reading of mRNA.

One notable aspect is TSTO's unique ability to define trimers, especially homomeric trimers (Supplementary Data 3). This has been challenging for bifunctional linkers due to the difficulty in identifying multimeric interactions. For instance, the trimer of nucleoside-disphosphate kinase B (NME2) was detected due to the identification of a TSTO tri-link connecting three identical sequences containing NME2:K100. Interestingly, NME2 is known to form homohexameric structures comprising two stacked homotrimers. When mapped to a high-resolution structure of NME2 (PDB:8PYW), the loop regions containing the K100 residues of each homotrimer were found to localize along the axis of the hexamer, within 9.5 Å of one another (Fig. 6a)[31]. Similarly, a homotri-link was identified for the 10 kDa mitochondrial heat shock protein (HSPE1) through its K54 and K56 residues, suggesting an oligomeric complex. Indeed, HSPE1 has been shown to form a homoheptameric ring within the human mitochondrial chaperonin 'football' complex (PDB: 4PJ1)[32], in which the longest distance spanned between any pair of HSPE:K54 or HSPE:K56 residues within neighboring triplets was 20 Å (Fig. 6b). In addition, homotrimeric TSTO cross-links were identified from proteins that are known to assemble into oligomeric complexes but currently lack high-resolution structures. For instance, ATPase family AAA domain-containing protein 3 A (ATAD3A) was shown to oligomerize through a specific residue (K262). Using AlphaFold3[33], we predicted the structure of an ATAD3A trimer at high confidence (90 > plDDT > 70) and mapped all possible K262 interactions satisfactorily (<18.5 Å) (Fig. 6c),

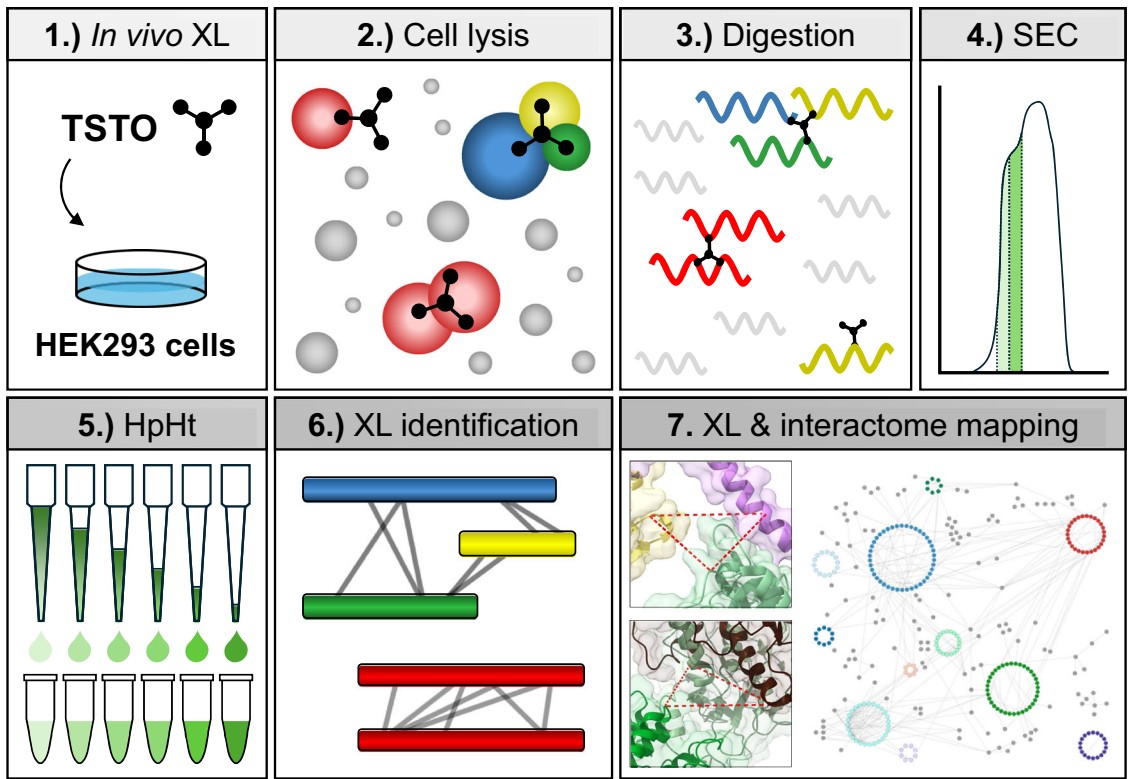

**Fig. 5 |** In vivo TSTO XL-MS analysis workflow.

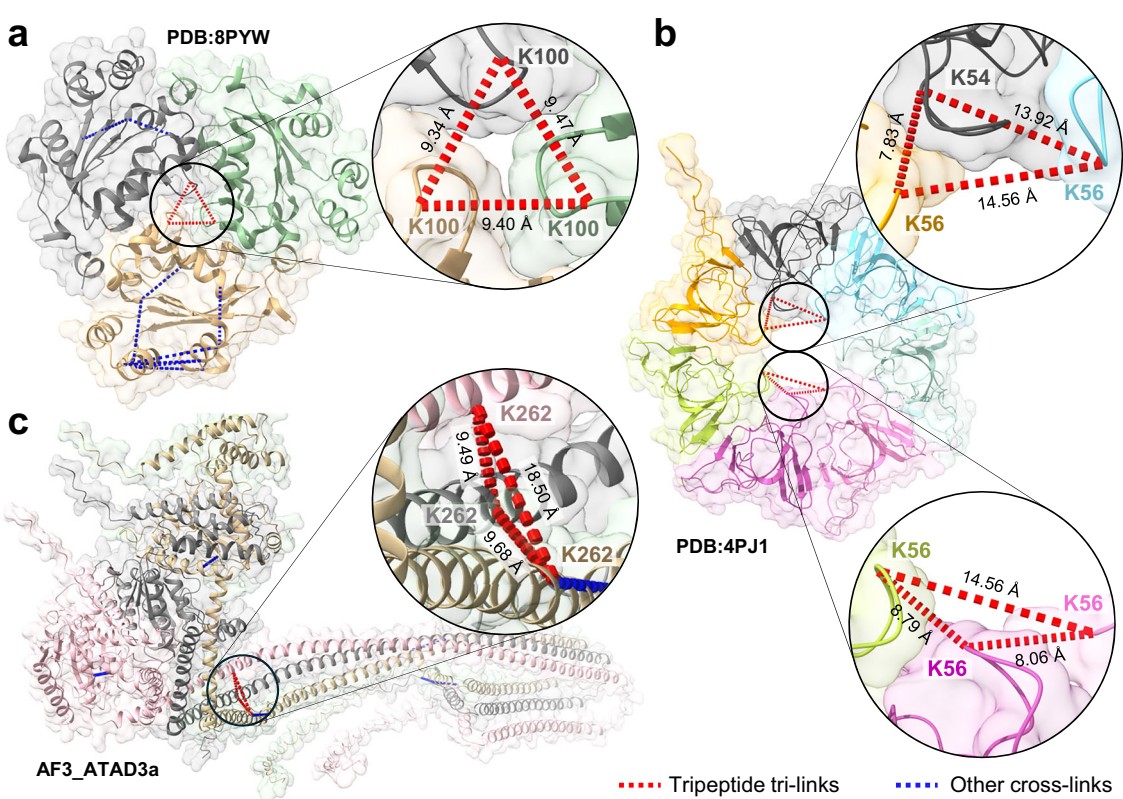

**Fig. 6 | Mapping of homotrimeric TSTO cross-links from in vivo XL-MS. a** Tri-link through K100 residues of three NME2 proteins, mapped to a high-resolution structure (PDB: 8PYW). **b** Trimeric cross-links involving residues K54 and K56 of three HSPE1 proteins mapped to a high-resolution structure (PDB:4PJ1). **c** Tri-link through K262 of three ATAD3a proteins, mapped to an AlphaFold3-generated model for an ATAD3a trimer.

exemplifying the capability of TSTO to facilitate structural modeling of protein oligomers.

### In Vivo TSTO Cross-linking of Mouse Heart Tissue

To illustrate the feasibility of TSTO in another biological context, we performed in-tissue cross-linking to identify cross-linked peptides using the same workflow for in-cell XL-MS as established above. Here, TSTO cross-linking of intact mouse hearts resulted in the identification of a total of 4823 CSMs using LC MS$^n$ (Supplementary Data 4). Among them, 191 unique trivalent and 861 unique bivalent cross-links were identified, providing a snapshot of protein interactions within the heart. The most abundant XL-PPIs involved proteins that are major structural and functional components of cardiac tissue (Supplementary Fig. 12), including actin, myosin, tropomyosin, and ATP synthase subunits. This observation correlates with the known role of these proteins in heart muscle function, particularly in the sarcomere—the contractile unit of muscle fibers.

To further explore protein interactions within the cardiac muscle, we generated an XL-PPI map focusing on the interactions of cardiac actins with myosins, tropomyosins, and troponins, all of which are key regulators of muscle contraction (Supplementary Fig. 13a). Myosin, a motor protein, binds actin and its function is regulated by tropomyosin and troponin. Tropomyosin, a coiled-coil protein, runs along actin filaments and controls myosin access to actin-binding sites, while the troponin complex (composed of troponin C, T, and I) binds actin and tropomyosin to modulate contraction by controlling myosin-binding site exposure. Consistent with these molecular functions, our cross-linking data revealed interactions among these proteins, with the most direct connections observed between myosin and actin, while tropomyosin and troponins were cross-linked with both actin and to each other (Supplementary Fig. 13a). Notably, trivalent cross-links were identified not only within myosins and tropomyosins but also between troponin and actin, as well as between actin and tropomyosin (Supplementary Data 4). In addition, we found that tropomyosin was cross-linked to an ATP synthase subunit. While this interaction is not typically associated with sarcomere function, it could suggest a potential regulatory link between contractile and energy-producing processes, possibly reflecting coordinated control of ATP synthesis and muscle contraction in cardiac cells. Further Gene Ontology (GO) enrichment analysis of heart-specific proteins captured by TSTO cross-linking revealed a significant representation of pathways associated with cellular energy metabolism, particularly those involved in ATP synthesis and mitochondrial function, which are critical for maintaining cardiac contractility and overall heart function (Supplementary Fig. 13b–d). Taken together, our results have shown the feasibility of TSTO for in-tissue cross-linking, expanding its applicability for different biological samples.

## Discussion

In this work, we have designed, synthesized, and characterized a novel trioxane-based, MS-cleavable, membrane-permeable homotrifunctional cross-linker, TSTO, to dissect multimeric protein interactions. This cross-linker enables simultaneous cross-linking of up to three proteins, allowing for more in-depth PPI analysis and providing additional restraints to advance structural analysis of protein assemblies. We have demonstrated that all types of TSTO cross-linked peptides display unique and predictable CID-induced fragmentation and can be unambiguously identified using LC MS$^n$ analysis. TSTO's distinct ability to concurrently release all three cross-link arms and leave an identical remnant on each cross-linked residue establishes it a brand-new class of MS-cleavable reagent. In addition, this distinctive feature minimizes the number of theoretical MS fragments corresponding to each peptide constituent, simplifying ion selection for subsequent MS$^3$ analysis. While MS$^2$-based data acquisitions have become popular for analyzing MS-cleavable dipeptide cross-links, we envision that MS$^3$-based

approaches would be preferred for the identification of tripeptide cross-links. The co-fragmentation of three peptides within a single spectrum heavily convolutes database searching, impeding identification of higher-order cross-linked species by MS$^2$-based approaches. Moreover, the central trioxane presents an innovative core structure for developing multifunctional MS-cleavable cross-linkers as one arm can be replaced with other functional groups—such as enrichment, isobaric, or reporter tags—to enable cross-link purification, quantitation, or further improve cross-link detection and identification. The NHS ester groups within TSTO can also be replaced by other reactive chemistries to target specific or non-specific amino acids[21,34–36], expanding the coverage of XL proteomes. As with TSTO, the cleavage of the trioxane within the mass spectrometer would release any additional functional groups, preventing their impact on cross-linked peptide identification. Thus, the development of TSTO presented here opens a new direction for designing diverse cross-linkers to further advance XL-MS technologies.

Compared to cross-linking reagents that target two residues, TSTO enhances the localization of interacting proteins by simultaneously targeting three residues, triangulating a multi-point attachment that provides greater spatial resolution and allows more precise mapping of protein interfaces and interaction sites. This is particularly useful for modeling multimeric interactions, especially for protein complexes that can exist in different compositional states. As bifunctional cross-linkers can only provide spatial restraints between two residues, it can be difficult to correctly assign them to individual compositional forms without reiterative modeling. Therefore, trivalent cross-links offer an additional restraint to help facilitate characterization of multi-protein interactions. As shown with cross-linking of the 26S proteasome, TSTO has identified trimeric interactions bridging subunits of different proteasome subcomplexes that accurately describe their positioning within the 26S holocomplex—such as those between the 19S lid and base, and the 19S base and 20S subcomplexes. In addition, the tri-links have accurately described the orientation of multi-subunit interactions within their respective subcomplexes, such as the localization of Dss1 to the outer 19S lid and the close proximities of solvent-accessible residues of the ATPase ring subunits lining the central pore of the 19S base. Moreover, integrative structural modeling of a proteasome subcomplex has demonstrated that trivalent cross-links are advantageous in providing additional spatial information to facilitate structure modeling with increased precision and accuracy. This analysis has established a solid foundation for us to better characterize the structural organization of cellular networks in the future.

Similarly, TSTO has identified trimeric interactions from in vivo XL-MS analyses, for instance, triangulating the positions of ribosome-interacting proteins relative to the larger 80S ribosomal machinery, as well as various subunit interfaces within and between its 40S and 60S subcomplexes. In addition to revealing the identities of heteromeric interactions, TSTO expands the capability of XL-MS to identify homomeric ones. While bifunctional cross-linkers can reveal homodimer formation by identifying cross-linked peptides with overlapping sequences, TSTO is capable of placing homodimeric interactions in the three-dimensional context of multiprotein assemblies by capturing a third residue, providing a clearer picture of how individual dimers are oriented as part of a larger protein complex. Furthermore, TSTO can unravel homotrimeric interactions—as confirmed for NME2 and HSPE1—which can be used in conjunction with structure prediction such as AlphaFold to model unknown structures.

While TSTO is highly effective for in-cell cross-linking at lower concentrations than bifunctional cross-linkers[20,26,37], the heterogeneity of TSTO cross-links appears to impact the number of unique PPIs obtained for proteome-wide analysis. Based on our data, 62% of all

inter-protein PPIs derived from trivalent cross-links were covered by bivalent ones. This is most likely attributed to the reduced number of unique interactions identified in the current dataset. However, it is noted that each interaction was identified with higher confidence as they were supported by an increased number of cross-links. In comparison to in-cell TSTO XL-MS analysis, the overall coverage of in-tissue XL-PPIs revealed by TSTO was less extensive, more likely due to the higher complexity and lower recovery of tissue proteins and cross-linked peptides. To enhance protein coverage and increase cross-link identification for proteome-wide experiments, fractionation of cross-linked protein complexes can be coupled with peptide separation to improve analysis sensitivity and dynamic range. Although cross-link enrichment through incorporation of affinity tags such as biotin, azide, or phosphonic acid have been successful with previous bifunctional cross-linkers, this approach is likely impractical for TSTO due to the central positioning of its trioxane functional group. However, recent developments in polyclonal antibodies that can recognize MS-cleavable cross-linkers[38] suggest that similar affinity purification strategies could be adapted to enrich TSTO cross-links and increase their MS detectability. Moreover, while MS[3]-based analysis allows accurate cross-link identification, it is less sensitive than MS[2] analysis. Thus, MS[2]-based workflows could be explored to further increase the sensitivity of TSTO cross-link identification. This will require software development to enable the identification of three cross-linked peptides from a single MS[2] spectrum. Due to the unique MS-cleavability of TSTO, we anticipate MS[2]-based identification of TSTO cross-links would be highly feasible compared to the identification of tri-links formed by conventional bifunctional cross-linkers. Given its unique ability to form trivalent cross-links and its robustness in cross-link identification, TSTO represents a promising cross-linker that can provide additional spatial restraints to significantly enhance our understanding of protein modules and their organization in complex biological systems.

Beyond improving structural characterization and expanding the scope of detectable interactions, the TSTO-based XL-MS platform can be coupled with isotope-based quantitative strategies to enable the interrogation of how multimeric protein interactions fluctuate in response to biological stimuli, disease states, or drug treatments, providing a deeper functional understanding of cellular organization. The isotopic labels can be introduced into TSTO cross-links by SILAC labeling of lysines[39], chemical labeling of cross-linked peptides with isobaric reagents (e.g., TMT)[40], or coding isotopic labels in the linker design.

In summary, we have successfully established and employed the TSTO-based XL-MS platform to capture trimeric interactions of protein complexes and cellular networks from intact cells and tissues, yielding structural details that cannot be easily obtained by existing reagents. The information obtained can be combined with AlphaFold prediction and/or integrative structural modeling to enhance the characterization of cellular networks in future studies. Therefore, the TSTO-based XL-MS platform represents a highly promising approach for advancing XL-MS technology towards systems structural biology in vivo.

## Materials and methods

### Ethical statement
Animal experiments were performed according to the protocol approved by the Institutional Animal Care and Use Committee (IACUC) of the University of California, Irvine. Heart tissue used in the experiment were obtained from adult (20-week-old) C57BL/6 wildtype mice. Mice were maintained at constant temperature (23 °C) and humidity, and housed at 12 h light/12 h dark cycles and with free access to food and water.

### Experimental design
This study did not involve randomization or blinding; no data or mice were excluded from the analysis.

### TSTO synthesis
Starting with delta-valerolactone, (1) acid-catalyzed ring opening in methanol provided a methyl ester, which was (2) converted to an aldehyde via Swern oxidation in an 85% yield over two steps. (3) The resulting aldehyde was treated with 10 mol% zinc-(II) chloride, affording a symmetric trioxane tris-ester. (4) Hydrolysis of the tris-ester afforded a tris-acid, which was (5) converted to TSTO by ester formation with NHS and TFAA.

### TSTO cross-linking of synthetic peptide Ac-SR8
Synthetic peptide Ac-SR8 was cross-linked at a concentration of 1 mM with 1 mM TSTO in anhydrous DMSO or pH 7.4 phosphate buffered saline. The reaction was carried out at RT with gentle vortexing, with aliquots of the original reaction volume being removed and quenched with ammonium bicarbonate at 5, 15, 30, and 60 min. After dilution by 1:500, the resulting cross-linked peptide mixtures were directly injected for MS analyses.

### TSTO cross-linking of BSA
BSA was cross-linked at 20 mM with 1 mM TSTO in pH 7.4 phosphate-buffered saline. The reaction was carried out for 1 hr at RT, followed by quenching with ammonium bicarbonate. The protein was then reduced with tris(2-carboxyethyl)phosphine (TCEP) for 30 min at RT and alkylated with iodoacetamide in the dark at RT for 30 min. Cross-linked proteins were then digested in 8 M urea buffer using LysC for 4 hrs at 37 °C, followed by trypsin digestion at 37 °C overnight after diluting the urea concentration to <1.5 M. The resulting peptide mixtures were extracted and desalted using C18 tips (Agilent) prior to MS analyses.

### Affinity purification and TSTO cross-linking of human 26S proteasomes
A stable 293[HBTH-Rpn1] cell line[41] was first grown to confluency. After native cell lysis, human 26S proteasomes were purified from the clarified lysate by binding to streptavidin–sepharose resin. Bead-bound proteasomes were cross-linked on-bead in PBS buffer (pH 7.5) with 0.75 mM TSTO for 1 h at room temperature. After quenching the cross-linking reaction using ammonium bicarbonate, the proteins were reduced with TCEP for 30 min at RT and alkylated with iodoacetamide in the dark at RT for 30 min. Cross-linked proteins were then digested in 8 M urea buffer using LysC for 4 hrs at 37 °C, followed by trypsin digestion at 37 °C overnight after diluting the urea concentration to <1.5 M. The resulting peptide mixtures were extracted and desalted using C18 tips (Agilent) prior to MS analyses.

### Optimization of in vivo TSTO cross-linking by immunoblot analysis
A stable 293[HBTH-CSN2] cell line[42] was first grown to confluency, washed with PBS, and then gently pelleted. To determine the optimal cross-linking conditions, intact cells were cross-linked at various cross-linking concentrations ranging from 0.5 to 3 mM, at room temperature or 37 °C. Clarified lysates from each condition were separated by SDS-PAGE and transferred onto a PVDF membrane and stained using amido black. After rinsing off the dye and blocking using 5% milk in TBST, the membrane-bound proteins were incubated with streptavidin-HRP to monitor the oligomerization of HBTH-CSN2 in response to cross-linking conditions. Based on these results, in vivo cross-linking of intact HEK293 cells was performed at 1 mM.

### In vivo TSTO cross-linking of human cells
Intact 293[HBTH-CSN2] cells were cross-linked using 1 mM TSTO in PBS buffer (pH 7.4) for 1 h with rotation at room temperature. Afterwards, the cross-linking reaction was quenched using excess ammonium bicarbonate (50 mM) for 10 min. Cells were spun down, washed again with PBS, and needle-lysed on ice in denaturing buffer (8 M urea,

50 mM Tris-HCl pH 7.5). Lysate was clarified by centrifugation at $21,000 \times g$ for 15 min in 4 °C, and the resulting supernatant was transferred to EMD Millipore 30,000 NMWL Microcon centrifugal tubes for FASP digestion similarly as described[43]. Briefly, proteins atop the filter were reduced using 10 mM TCEP for 20 min at RT, alkylated using 20 mM iodoacetamide in the dark at RT for 20 min, and then digested in 8 M urea buffer using LysC for 4 h at 37 °C followed by trypsin digestion at 37 °C overnight after urea dilution to 1.5 M. The resulting cross-linked peptide mixtures were then spun through the filter and desalted using Waters C18 Sep-Pak cartridges.

### In vivo TSTO cross-linking of mouse heart tissue

Freshly excised mouse hearts were sliced into 1-2 mm cubes and washed in cold PBS to remove blood. The cubed cardiac tissue was incubated in 3 mL of 1 mM TSTO in PBS for 1 hr at RT with rotation. The cross-linking reaction was then quenched using 20 mM ammonium bicarbonate for an additional 15 min rotation at RT. The tubes were then gently centrifuged, and the tissues were washed using cold PBS twice. The tissues were then flash frozen and manually cryopulverized before lysis using a BioRuptor in denaturing buffer (8 M urea, 50 mM Tris-HCl pH 7.5). 5 cycles consisting of 30 s sonication followed by 30 s rest were used to lyse the cryopulverized tissue. Lysate was clarified by centrifugation at $21,000 \times g$ for 15 min in 4 °C, and the resulting supernatant was transferred to EMD Millipore 30,000 NMWL Microcon centrifugal tubes for FASP digestion as described above. Following digestion with LysC and trypsin, the resulting cross-linked peptide mixtures were then spun through the filter and desalted using Waters C18 Sep-Pak cartridges.

### SEC-HpHt enrichment of cross-linked peptides.

Peptide separation by SEC was performed similarly as described[44]. Briefly, dried peptides were reconstituted in SEC mobile phase (0.1% formic acid and 30% ACN) and separated on a Superdex Peptide PC 3.2/30 column ($300 \times 3.2$ mm) at a flow rate of 50 μL/min, monitored at 215, 254 and 280 nm UV absorbance. Two-minute fractions were collected, and only fractions 24 and 26 containing the most cross-linked peptides were collected. SEC-separated fractions were then vacuum dried and resuspended in ammonium water (pH 10). For high-pH separation, HpHt tips were prepared as described[19]. Pipette tips (200 μL) were first blocked with a layer of C8 membrane (Empore 3 M), then filled with 5 mg of C18 solid phase (3 μm, Durashell, Phenomenex). The tips were then washed sequentially with 90 μL of methanol, 90 μL of ACN and 90 μL of ammonia water (pH 10). After loading onto the tip, the peptides were washed once with 90 μL ammonia water and eluted using increasing percentage of ACN in ammonia water (6, 9, 12, 15, 18, 21, 25, 30, 35, and 50%). The 25, 30, 35 and 50% fractions were then combined with 6, 9, 12 and 21% fractions, respectively. The final SEC-HpHt fractions were vacuum dried and stored at − 80 °C before MS analysis.

### LC-MS$^n$ analysis.

LC-MS$^n$ analysis of cross-linked peptides was performed using an UltiMate 3000 UPLC (Thermo Fisher Scientific) liquid chromatograph coupled online to an Orbitrap Fusion Lumos mass spectrometer (Thermo Fisher Scientific). Peptides were separated by reverse-phase on a 50 cm × 75 μm I.D. Acclaim® PepMap RSLC column using gradients of 4% to 25% acetonitrile at a flow rate of 300 nL/min (solvent A: 100% $H_2O$, 0.1% formic acid; solvent B: 100% acetonitrile, 0.1% formic acid) prior to MS$^n$ analysis. For each MS$^n$ acquisition, duty cycles consisted of one full Fourier transform scan mass spectrum (375–1500 m/z, resolution of 60,000 at m/z 400) followed by data-dependent MS$^2$ and MS$^3$ acquired at top speed in the Orbitrap and linear ion trap, respectively. Ions detected in MS$^1$ with 4 + or greater charge were selected and subjected to CID fragmentation (NCE 23%) in MS$^2$, and resulting ions were detected in the Orbitrap (resolution 30,000). Ions observed in MS$^2$ spectra with charge 2 + or greater were

selected and fragmented in MS$^3$ using CID (NCE 35%) and detected in the linear ion trap in 'Rapid' mode. Ions selected for MS$^3$ were either based on abundance (top 4 or 5 most intense ions in MS$^2$) or targeted based on doublets with mass difference pairs (Δ = 18.02 Da) corresponding to cross-linker remnant moiety water loss. For 26S proteasome cross-links, each acquisition was 200 min and ion selection for MS$^3$ was based on top intensity. For SEC-separated F24 fractions, the entire HpHt fraction was injected; for F26 fractions, half of the HpHt fraction was injected for each LC-MS$^n$ analysis. Each acquisition was 120 or 150 min, and both top intensity and mass difference-targeted methods were used for selecting ions for MS$^3$ analysis.

### Identification of TSTO Cross-links by MS$^n$

Spectrometric data were extracted from.raw files using PAVA[45]. Extracted MS$^3$ spectra were subjected to protein database searching via Batch-Tag within a developmental version of Protein Prospector (v. 6.3.5, University of California, San Francisco) against a randomly concatenated SwissProt database consisting of 20,418 human proteins and their corresponding decoys. Mass tolerances for parent ions and fragment ions were set as ± 15 ppm and 0.6 Da, respectively. Trypsin was set as the enzyme with three maximum missed cleavages allowed. Cysteine carbamidomethylation was selected as a constant modification, while protein N-terminal acetylation, methionine oxidation, and N-terminal conversion of glutamine to pyroglutamic acid were selected as variable modifications. Two additional defined variable modifications on uncleaved lysines and free protein N-termini were selected: AR ($C_5H_6O_2$, 98.04 Da) and AR* ($C_5H_4O$, 80.03 Da), corresponding to remnant moieties for cleaved TSTO. MS$^n$ data (monoisotopic masses and charges of parent ions and corresponding fragment ions and MS$^3$ database search results were integrated via in-house software xl-Tools[39] to automatically generate, summarize and validate identified cross-linked peptide pairs. Experimental FDR calculated using a target-decoy approach was determined to be 0.04% at the cross-link level. Using a minimum peptide length of 5 residues for individual cross-linked peptides, the separation of inter/intra FDR calculation resulted in experimental FDRs of 0.2% and 0.0%, respectively.

### Integrative modeling using TSTO data

Integrative modeling was used to demonstrate the effectiveness of the TSTO cross-linker in structural modeling by computing the structure of the 26S human proteasome base subcomplex (subunits Rpt1–Rpt6 and Rpn2[15]) based on data from trifunctional and bifunctional cross-linkers. Our protocol proceeded through the standard four stages:

*(1) Gathering information:* Input information consisted of AlphaFold2-predicted structures[46] for the subunits (Rpt1–Rpt6 and Rpn2), 642 TSTO cross-links (XLs) identified in this study, 281 DSSO cross-links reported by Yu et al.[16], and synthetically generated cross-linking datasets. Synthetic cross-linking datasets were generated based on the cryo-EM structure of the human 26S proteasome (PDB ID: 5GJR[47]). Cross-linking probabilities were assigned to lysine residue pairs with Cα–Cα distances below 30 Å using a skewed normal distribution, with parameters (location: 8.0, scale: 14.8, skewness: 7.6) derived from fitting the experimental distance distribution of trivalent cross-links. Monte Carlo sampling, guided by the assigned probabilities and the Metropolis acceptance criterion, was used to select residue pairs and triplets for constructing the synthetic datasets.

*(2) Representing the subunits and translating data into spatial restraints:* Relying on the AlphaFold2 predictions of the proteasome base subcomplex, we represented each subunit as a rigid body using a 1-residue-per-bead representation. Spatial restraints for cross-linked residues were applied as upper bounds on the distances between residue pairs. TSTO trivalent cross-links were represented by a set of three distance restraints between all residue pairs,

ensuring that distances spanned by all residue pairs are simultaneously within the cross-linker distance threshold. TSTO and DSSO bivalent cross-links were modeled using a single upper-bound distance restraint. Furthermore, excluded volume and sequence connectivity restraints were imposed on all components.

*(3) Sampling:* Structural models were computed using Replica Exchange Gibbs sampling, based on the Metropolis Monte Carlo (MC) algorithm. Each MC step consisted of a series of random transformations (i.e., rotations and/or translations) of the rigid bodies.

(4) Analysis and validation of the structural models of the 26 s proteasome followed the previously published five steps[46,48,49].

To benchmark the four-stage protocol described here and assess the utility of the TSTO cross-linker for integrative structure modeling, we computed the distribution of the accuracy for the structural ensembles obtained by integrative modeling. The accuracy is defined as the mean of Cα RMSD between the EM structure and each of the structures in the ensemble. In addition, we computed the precision of each ensemble of models; the precision is defined as the average RMSD between all solutions in the ensemble.

The integrative structure modeling protocol [i.e., stages (ii), (iii), and (iv)] was scripted using the *Python Modeling Interface* (PMI) package, a library for modeling macromolecular complexes based on our open-source *Integrative Modeling Platform* (IMP) package version 2.20. All input files, scripts, and output files are available at [GitHub link].

### Reporting summary

Further information on research design is available in the Nature Portfolio Reporting Summary linked to this article.

## Data availability

Unless otherwise stated, all data supporting the results of this study can be found in the article, supplementary, and source data files. The mass spectrometry data generated in this study have been deposited to the ProteomeXchange Consortium via the PRIDE partner repository with the dataset identifiers PXD054551 and PXD063825. Input files, scripts and data generated during integrative structure modeling are archived at Zenodo with accession code 15367945. In addition, the following published structures were used for cross-link mapping or structural modeling: 4F5S 7QY7 5GJR 8PYW 4PJ1 Source data are provided in this paper.

## Code availability

The *Python Modeling Interface* (PMI) package is available at [GitHub link].

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

## Acknowledgements

We wish to thank Prof. A.L. Burlingame, Drs. Peter Baker and Robert Chalkley for their support of Protein Prospector, and the Huang lab members, especially Paul Morenkov and Fenglong Jiao, for their help. This work was supported by National Institutes of Health grants R35GM145249 to L.H., R01CA290875 to A.W. and L.H., R35GM151256 to I.E., and R01DK136940 and R01DK121146 to Q.Y.

## Author contributions

L.H. conceived the study and directed the research. C.Y., X.W., E.N., S.R., and L.H. designed the experiments. C.Y. performed XL-MS experiments, data acquisition and analyses. X.W. purified and cross-linked proteasome complexes, E.N. and S.R. performed cross-linker synthesis, S.G.B. and I.E. performed integrative structural modeling and AlphaFold prediction and wrote the structural modeling section. X.S. and Q.Y. performed mouse dissections for tissue extraction. C.Y. and L.H. wrote the manuscript with contributions from other authors.

## Competing interests

The authors declare no competing interests.
