## [Peer Review file · Nature Communications]

Trioxane-based MS-cleavable Cross-linking Mass Spectrometry for Profiling Multimeric Interactions of Cellular Networks

Corresponding Author: Dr Lan Huang

Version 0:

Reviewer comments:

Reviewer #1

(Remarks to the Author)

The work presents an interesting and useful addition to the cross-linking mass spectrometry toolbox. However, one of the major examples used in the manuscript to demonstrate the power of the new reagent, the positioning of transient subunit Dss1 (SEM1) in the 26S proteasome lid, is not presented in a way supporting appreciation of the work. It is not exactly true that the positioning of Dss1 in the human 26S proteasome lid has not been defined. Dss1 is present in the human 26S proteasome structures solved by cryoEM, for example in the PDB: 7QY7 cited by the Authors and also in PDB: 5L4K (2016). Both structures show Dss1 in proximity of Rpn6 and Rpn3. The yeast 26S proteasome CryoEM structures show Dss1 positioning as well, in proximity of Rpn7 and Rpn3 (PDB: 5MPD, 2017), as cited by the Authors (ref. 22). The cross-linking data presented in the manuscript suggest proximity of Rpn3, Rpn6 and Rpn7 in the human 26S proteasome lid. However, in 7QY7 and also in 5L4K the Rpn7 is positioned far away from Dss1. Rpn7 may interact with Dss1 in the yeast proteasome lid, where in turn Rpn6 is far away from Dss1. The reported cross-links with three Rpn subunits seem to combine yeast and human interactome. However, the discrepancy with human structures 7QY7 and 5L4K is not noted and not discussed. This needs to be addressed.

Minor:

- Ref. 8 does not tell about Dss1 and only cites 2017 work about the 26S.
- Figure S3 caption: there is a typo in PDB structure name: should be 7QY7, not 7QX7; the latter is not proteasome.

Reviewer #2

(Remarks to the Author)

Authors (Ye C. et al,...) have designed, synthesized, and characterized a novel trioxane-based, MS-cleavable, membrane-permeable homo-trifunctional cross-linker, TSTO, to dissect multimeric protein interactions. With this new designed crosslinker authors can simultaneous cross-linking of up to three proteins (peptides), allowing for more in-depth PPI analysis and providing additional restraints to advance XL-MS based structural analysis of protein assemblies.

First proof of concept analysis was done with human 26 S proteasomes which shows up promising results. Further authors demonstrated that all types of TSTO cross-linked peptides display unique and predictable CID-induced fragmentation and can be unambiguously identified using LC MSn analysis.

From my own experience I know how difficult it is to design and synthesize new cross-linkers. There are many failed attempts that are unfortunately not published. Unfortunately, that's what happened to us. That's why I admire the group around Lan Huang so much, who have proven with this work that they deliver world-class work. This work (development of TSTO) is a completely new concept which could be published in Nature Com. or higher ranked journal. I am also pretty sure that there will be several follow-up publications based on this work.

In my opinion, this manuscript is very well written, and I think there are a few small things to improve. Therefore, I would consider it a minor review overall.

Major items.

Please separate inter/intra FDR calculation and minimal peptide length on FDR as published by our lab (Matzinger et al.

Nat.Com 2022).

Just wonder if there is a possibility to validate results with another software package like MS-Anika?
The dynamic area of the method has not been mentioned here, but would at least be worthy of an supplemental figure.

Minor items: Please name the manufacturer of 30,000 NMWL Microcon centrifugal tubes
LC-MS settings: Unclear how much volume or μg of digested peptides were injected
Please have the method section re-evaluated by a person not involved in the project to correct any ambiguities.
The protocol includes multiple separation steps like SEC or high-pH enrichment. Just wonder if there is a possibility to quantify crosslinked peptides without missing values?

Reviewer #3

(Remarks to the Author)

The manuscript describes application of a new cleavable cross linking agent TSTO equipped with three identical reactive groups toward lysines. The power of the TSTO is presented by the detection of residues connectivity within the 26S proteasome particle, ribosomes and in cultured cell extracts. The data as presented is impressive and it is possible that the linker will find broad applications. Indeed, spatial distribution of lysines in proteins makes them target of choice for many decades, so any additional information on their interactions is highly anticipated. This reviewer did not follow MS analysis peptide by peptide assuming the correct sequence assignments. However, the manuscript would benefit from inclusion of additional explanations.

1. Chemical characterization of TSTO is missing.
2. There is no word how specific is the crosslinker (other potential targets?) and potential side reactions.
3. No control reactions reported that would show no cross links when lysines are modified or mutated.
4. The 26S particles were affinity purified with Rpn1 but their quasi stability has not been addressed. The concern here is that many of potential links are missing or even spurious when a complex and dynamic protein as proteasome is tested with crosslinkers.
5. Trifunctional crosslinker like TSTO shows obvious limitations arising from the necessity to simultaneously react with three residues placed at a proper distance. It has not been addressed how reaction time with TSTO affects the detected links and if protein dynamics could be deduced from the detected links (allosteric forms?).
6. Although the Authors claim that the trivalent cross linker is better than tetra or di-valent but formal proof of that is absent.
7. It is surprising that in an XL proteome of HEK293 cells 1512 proteins were found but only 1242 interactions were revealed. Is it an indication of the limitation of TSTO reactivity?
8. The results presented in Figure S4 need more complete association with the stipulated cell permeability. The selection of 1mM TSTO also needs better explanation. The immunoblot shows monomer of casein and after the crosslinking only a band of very large aggregates (?) but no other oligomers.
9. I would recommend inclusion of the major findings in the abstract. Currently, we are only treated to an introduction of TSTO with a very general advertisement of its power.
10. Very minor: in Fig 2A labels of subunit shifted.
11. Very minor: the mw ladder band representing 50kDa is out of the position – the distance between 50 and 60 kDa is much larger than between 40 and 50 kDa. Is this a gradient gel?

Reviewer #4

(Remarks to the Author)

C. Yu and co-authors' manuscript relates the development of new generation of cross-linkers to allow interactomics approach based on an innovative homo tri-functionalized linker allowing trimeric cross-links. The trioxane derivative MS-cleavable cross linker allows an MS3 clear identification of targeted fragments after digestion. The authors define properly this very interesting tool, firstly by detailing the chemical synthesis of the TSTO XL-linker. Then, they highlight the tagging effect through the detection of specific fragment ions signature by MS2 generation of α , β , γ modified peptides coupled with their * ions corresponding to the loss of H₂O. This "featured" signature may potentially not be sufficiently unambiguous and should be more discussed and illustrated. The authors present a very interesting proof of concept using the model of 26S Proteasomes interactomic study with the identification of 808 unique CSMs and 41% of tri-linked species. However, Authors should more particularly describe the gain of information compared to bifunctional XL-MS platforms. Although the results are relevant and clearly reported, the present strategy may not be sensitive enough to be used in an endogeneous context, because of an obvious loss of sensitivity due to the multiplicity and diversity of signals. The authors don't illustrate the possibility of obtaining monomeric species, which represents a fourth type of "cross-link". Finally, exploration of the multiplex potential of this tool could be of primary importance. The discussion section should be expanded on this point, and the results should be demonstrated more thoroughly, emphasizing the potential of this technique in real in vivo conditions, as HEK293 cell culture may not be representative of an in vivo model.

Version 1:

Reviewer comments:

Reviewer #1

(Remarks to the Author)

All concerns have been addressed. Thank you very much.

Reviewer #2

(Remarks to the Author)

The authors have done a great job and answered all my comments and suggestions to my complete satisfaction. The quality of the publication has been significantly improved and can now be published as it is.

I hope to read many more exciting articles from the group and wish them all the best in the current difficult circumstances (funding).

Reviewer #3

(Remarks to the Author)

The Authors of this revised version of the manuscript addressed all major concerns of the reviewer and satisfactorily answered reviewer's questions.

Reviewer #4

(Remarks to the Author)

Dear Authors,

Thank you for addressing every concern with such accuracy and quality. Your work has generated new, very interesting, and high-quality data.

Detailed Responses to Reviewers' Comments:

Reviewer #1

Q1: *The work presents an interesting and useful addition to the cross-linking mass spectrometry toolbox. However, one of the major examples used in the manuscript to demonstrate the power of the new reagent, the positioning of transient subunit Dss1 (SEM1) in the 26S proteasome lid, is not presented in a way supporting appreciation of the work. It is not exactly true that the positioning of Dss1 in the human 26S proteasome lid has not been defined. Dss1 is present in the human 26S proteasome structures solved by cryoEM, for example in the PDB: 7QY7 cited by the Authors and also in PDB: 5L4K (2016). Both structures show Dss1 in proximity of Rpn6 and Rpn3. The yeast 26S proteasome CryoEM structures show Dss1 positioning as well, in proximity of Rpn7 and Rpn3 (PDB: 5MPD, 2017), as cited by the Authors (ref. 22). The cross-linking data presented in the manuscript suggest proximity of Rpn3, Rpn6 and Rpn7 in the human 26S proteasome lid. However, in 7QY7 and also in 5L4K the Rpn7 is positioned far away from Dss1. Rpn7 may interact with Dss1 in the yeast proteasome lid, where in turn Rpn6 is far away from Dss1. The reported cross-links with three Rpn subunits seem to combine yeast and human interactome. However, the discrepancy with human structures 7QY7 and 5L4K is not noted and not discussed. This needs to be addressed.*

A1: We thank the reviewer for pointing this out and apologize for the confusion. In addition to Rpn3 and Rpn6, we have identified three cross-links between Rpn7 and Dss1 (Rpn7:K284-Dss1:K66, Rpn7:K349-Dss1:K62, and Rpn7:K349-Dss1:K66) using TSTO cross-linking. While the reviewer suggests that Rpn7 is positioned far away from Dss1 in published structures of the human 26S proteasome (PDB: 5L4K (Schweitzer et al, PNAS, 2016) and 7QY7 (Lee et al., Mol Cell, 2023)), it is apparent that certain regions of Rpn7 are proximal to Dss1. Since the 5L4K structure represents a structure of the 19S, we decided to select two human 26S structures for mapping the Rpn7-Dss1 cross-links (PDB:7QY7 and PDB: 6MSB (Dong et al., Nature, 2019)) The latter was added in Figure S5. Based on the spacer arm length of TSTO (~14 Å), the C α -C α distances between Rpn7 and Dss1 cross-linked lysines are expected to be ≤ 35 Å. When mapped, all Rpn7-Dss1 cross-links corresponded to C α -C α distances ≤ 35 Å (ranging from 21.4 Å to 25.9Å) (Figure S5), thus confirming the proximity of Dss1 to Rpn7. To clarify this, we have revised the text as follows:

“Moreover, TSTO cross-links placed a small proteasome subunit Dss1 in proximity to Rpn3, Rpn6 and Rpn7 within the human 26S proteasome (Figures 3E, S5), which has not been reported by previous XL-MS analyses (Wang, *MCP*, 2017; Yu, *Anal Chem*, 2022). Nonetheless, these interactions were supported by the two human 26S structures (PDB: 5MSB (Dong, *Nature*, 2019) and 7QY7 (Lee, *Mol Cell*, 2023)) as all cross-links between Dss1 and Rpn3, Rpn6 and Rpn7 have C α -C α distances ≤ 26 Å, falling below the distance threshold (Figure 3E, S5)”

Q2: *Ref. 8 does not tell about Dss1 and only cites 2017 work about the 26S.*

A2: Thank you, we have corrected the references for the two high-resolution structures of the human 26S proteasome containing Dss1: Dong et al., *Nature*, 2019; and Lee et al., *Mol Cell*, 2023, as mentioned above.

Q3: *Figure S3 caption: there is a typo in PDB structure name: should be 7QY7, not 7QX7; the latter is not proteasome.*

A3: Thank you, we have corrected the figure caption appropriately.

Reviewer #2

Q1. *Please separate inter/intra FDR calculation and minimal peptide length on FDR as published by our lab (Matzinger et al. Nat.Com 2022).*

A1: Thanks for the suggestion and we have implemented this information accordingly in supplemental information.

In total, TSTO identified 7444 unique CSMs, of which 1465 were inter-protein and 5980 were intra-protein cross-links. Of these, three inter-protein CSMs were attributed as decoys, resulting in an overall experimental FDR of 0.04%. Separation of inter/intra FDR calculation resulted in experimental FDRs of 0.2% and 0.0%, respectively, and remaining well below the typically accepted 1% FDR threshold for cross-link identification.

Next, we determined the distribution of cross-linked peptide IDs based on the shortest length peptide within each cross-link:

Figure 1. Histogram showing the distribution of cross-linked peptides against length of the shortest peptide per cross-link.

Three decoy-containing cross-links were identified with minimum peptide length 5, 6, and 7. Thus, we calculated the resulting inter-protein FDRs when setting the minimum peptide length for cross-link identification as 5, 6, and 7 residues, yielding experimental FDRs of 0.04%, 0.03%, and

0.02%, respectively. Therefore, we feel that it is reasonable to maintain 5 as the minimum peptide length for our current dataset. Although Matzinger et al. have shown a clear benefit to separating inter/intra FDR calculation, as well as defining a minimum peptide length for FDR analysis for MS²-based workflows, there is a fundamental difference in database searching between MS³- and MS²-based workflows. MS³ search aims to identify single sequences, whereas MS² search attempts to identify two sequences simultaneously. In addition, the integration of MS¹, MS² and MS³ data provides additional filtering and stringency for cross-link identification. Thus, in contrast to MS²-based analysis, separation of inter/intra FDR calculation and minimal peptide length had a nominal effect on experimental FDR for MS³-based analysis.

We have added additional details regarding minimum peptide length and inter/intra FDR calculations in the methods section:

“Experimental FDR calculated using a target-decoy approach was determined to be 0.04% at the cross-link level. Using a minimum peptide length of 5 residues for individual cross-linked peptides, separation of inter/intra FDR calculation resulted in experimental FDRs of 0.2% and 0.0%, respectively.”

Q2: Just wonder if there is a possibility to validate results with another software package like MS-Anika?

A2: We thank the reviewer for their suggestion and have attempted to use both MS-Annika and XlinkX within the Proteome Discoverer suite. As these software packages are designed only for identifying bivalent cross-links, we employed them to search for TSTO dipeptide cross-links. However, one of the major issues with defining cleavage of TSTO within Proteome Discoverer is that cross-linker cleavage is required to yield two nonequivalent arms (e.g. short and long modifications, such as alkene and thiol moieties for sulfoxide-based cross-linkers). In the case of TSTO, one can only define both AR and AR*, despite AR* being the dominantly identified fragment. As a result, while both MS-Annika and XlinkX were able to identify *some* cross-links, most matches were based primarily on cross-link fragments and backbone cleavage within MS² spectra, as opposed to utilizing MS³ scans to match peptide backbone cleavage ions. This resulted in significantly less identifications compared to our current XL-MS workflow. We have reached out with this request to the Proteome Discoverer developers and have been told that defining cross-linking reagent cleavage resulting in equal mass arms will be enabled in a future release. Therefore, these software are not currently suited for our current study.

Q3: The dynamic area of the method has not been mentioned here, but would at least be worthy of a supplemental figure.

A3: We thank the reviewer for their suggestion and have included an additional supplemental figure (Figure S10) and text in the section ‘In Vivo TSTO Cross-linking of the HEK293 Cells’ to discuss the dynamic range of proteins capture by TSTO cross-linking as follows:

“Finally, to estimate the dynamic range of the XL proteome captured by TSTO cross-linking, we plotted the abundance distribution of identified cross-linked proteins based on their copy numbers as determined by shotgun proteomics (Bekker-Jensen, *Cell Syst*, 2017) (Figure S10). Compared to the MS-proteome, the TSTO XL-proteome was shifted towards higher abundance proteins,

however, it is comparable to previous proteome-wide XL-MS studies (Wheat, *PNAS*, 2021; Jiang, *Angew Chemie*, 2022; Jiao, *Anal Chem*, 2022). Similarly, the results indicate that TSTO is capable of targeting cellular proteins across all cellular compartments and capturing interactions among proteins spanning five orders of magnitude.”

Figure S10. Comparison of protein abundances across the TSTO XL-proteome, MS-proteome and several published XL-proteomes. Protein abundance distribution of the human cell MS proteome determined by shotgun proteomics⁶ (red) compared to selected XL-proteomes (alkyne-A-DSBSO in vivo dataset from Wheat et al.⁵ (blue), tBu-PhoX in vivo dataset from Jiang et al.⁷ (purple), DSSO in vitro dataset from Jiao et al.⁸ (yellow), and TSTO in vivo dataset from this work (grey)).

Q4: Please name the manufacturer of 30,000 NMWL Microcon centrifugal tubes
 LC-MS settings: Unclear how much volume or μg of digested peptides were injected
 Please have the method section re-evaluated by a person not involved in the project to correct any ambiguities.

A4: Thank you for the suggestion. We have provided additional details to the method section to clarify potential ambiguities:

“Lysate was clarified by centrifugation at 21,000g for 15 min in 4 °C and the resulting supernatant was transferred to EMD Millipore 30,000 NMWL Microcon centrifugal tubes for FASP digestion”

“For SEC-separated F24 fractions, the entire HpHt fraction was injected; for F26 fractions half of the HpHt fraction was injected for each LC-MSⁿ analysis.”

Q5: The protocol includes multiple separation steps like SEC or high-pH enrichment. Just wonder if there is a possibility to quantify crosslinked peptides without missing values?

A5. Similar to quantitative analysis of linear peptides in shotgun proteomics, label-free based quantitation of cross-linked peptides in complex samples with multiple separation steps can be error-prone due to potential variations in separation and potential loss of low abundance peptides. To improve quantitation accuracy and prevent missing values, we have previously developed the QMIX (Quantitation of Multiplexed Isobaric-labeled cross (X)-linked peptides) method, a strategy for quantifying cross-linked peptides using isobaric mass tags (Yu, C. et al, Anal Chem. 2016). The same strategy can be incorporated with TSTO XL for multiple cross-linked peptides. After incorporating isobaric mass tags (e.g. TMT), cross-linked peptides can be enriched and separated for multiplexed quantitation to eliminate missing values. Potentially, isotopic labels can be coded into TSTO linker to enable multiple step separation to minimize variance and improve the quantitative analysis. This has been incorporated in the discussion:

“Beyond improving structural characterization and expanding the scope of detectable interactions, TSTO-based XL-MS platform can be coupled with isotope-based quantitative strategies to enable the interrogation of how multimeric protein interactions fluctuate in response to biological stimuli, disease states, or drug treatments, providing a deeper functional understanding of cellular organization. The isotopic labels can be introduced into TSTO cross-links by SILAC labeling of lysines (Yu, 2019), chemical labeling of cross-linked peptides with isobaric reagents (e.g. TMT)(Yu, 2016), or coding isotopic labels in the linker design.”

Reviewer #3:

***Q1:** This reviewer did not follow MS analysis peptide by peptide assuming the correct sequence assignments. Chemical characterization of TSTO is missing.*

A1: Thank you for the suggestion and we have added the details for TSTO characterization using a standard peptide SR8 and a model protein BSA in the sections titled ‘Characterization of TSTO Cross-linked Synthetic Peptide by MSⁿ Analysis’ and ‘Characterization of TSTO Cross-linked BSA by MSⁿ Analysis’:

“Characterization of TSTO Cross-linked Synthetic Peptide by MSⁿ Analysis

We first characterized TSTO cross-linking on the synthetic peptide Ac-SR8 (Ac-SAKAYEHR). Under our experimental conditions, three Ac-SR8 cross-linked products were detected: dead-end modified Ac-SR8 (α_{DN}), inter-linked Ac-SR8 homodimer [α - α], and Ac-SR8 homotrimer [α , α , α]. MS² analysis of dead-end modified Ac-SR8 (m/z 667.3163²⁺) yielded two dominant ions (m/z 542.2656²⁺, 551.2708²⁺) (Figure S1A). MS³ peptide fragment analysis determined these ions to be AR-modified Ac-SR8, with the lower mass ion corresponding to an AR moiety undergoing water loss (namely AR*), resulting in the detection of an ion doublet (α_{AR} and α_{AR^*}) with mass difference (Δ) of 18.02 Da (Figure S1B-C). Two differently-charged species of inter-linked Ac-SR8 homodimer [α - α] were detected (m/z 773.7071³⁺, 580.5319⁴⁺), each fragmenting into dominant ions corresponding to α_{AR} and α_{AR^*} during MS² analysis (Figure S2A-B). Similarly, MS² analysis of the Ac-SR8 homotrimer tri-link [α , α , α] (m/z 826.6508⁴⁺) yielded

three dominant ions ($\alpha_{AR}^{*2+}/\alpha_{AR}^{2+}/\alpha_{AR}^{1+}$) (Figure S2C). While MS³ analyses of both α_{AR} and α_{AR}^* resulted in their unambiguous identification (Figure S1B-C), selecting AR*-modified peptides for sequencing would be preferred due to the AR moiety's propensity for dehydration.

Characterization of TSTO Cross-linked BSA by MSⁿ Analysis

To characterize TSTO cross-linking in proteins, we performed XL-MS analysis on the model protein BSA, focusing on TSTO inter-linked peptides. As a result, all three types (I-III) of TSTO inter-linked peptides were identified by LC MS^e, each displaying the characteristic MS² fragmentation as expected. This is illustrated by MSⁿ analyses of representative TSTO cross-linked BSA peptides (Figure S3). For a tripeptide tri-link [α , β , γ] (m/z 795.7120⁶⁺), its MS² analysis yielded three sets of dominant ions corresponding to $\alpha_{AR}/\alpha_{AR}^*$, β_{AR}/β_{AR}^* , and $\gamma_{AR}/\gamma_{AR}^*$ fragments (Figure S3A). As shown, MS³ analyses of the three cross-link fragments α_{AR}^* (m/z 638.3130²⁺), β_{AR}^* (m/z 773.8743⁺), and γ_{AR}^* (m/z 947.9347⁺) identified a tripeptide TSTO tri-link among BSA lysines K228, K374, and K498 (Figure S3B). MS² analysis of a dipeptide tri-link [α - β_2] (m/z 1054.7530⁴⁺) resulted in two sets of dominant ion species: $\alpha_{AR}/\alpha_{AR}^*$, and $\beta_{2AR}/\beta_{AR_AR^*}/\beta_{2AR}^*$ (Figure S3C). The detection of a fragment triplet with 18 Da increments indicates that peptide β carries two modified lysines, whereas peptide α only contains a single modified lysine. MS³ analyses of α_{AR}^* and β_{2AR}^* identified their sequences as ³⁷²LAK_{AR}*EYEATLEECCA³⁸⁶ and ⁴⁹⁰TPVSEK_{AR}*VTK_{AR}*CCTESLVNR⁵⁰⁷, respectively (Figure S2D), signifying a dipeptide tri-link [BSA:K374 - BSA:K495, K498]. Finally, for a dipeptide bi-link [α - β] (m/z 992.7017⁴⁺), MS² fragmentation produced two dominant ion pairs: $\alpha_{AR}/\alpha_{AR}^*$, and β_{AR}/β_{AR}^* (Figure S3E); MS³ analyses of α_{AR}^* and β_{AR}^* determined a cross-link between BSA:K117 and BSA:K489 (Figure S3F).

In total, 823 redundant cross-linked spectra matches (CSMs) were identified, corresponding to 167 unique ones (Table S1). Of these, 21 were tripeptide tri-links, 24 were dipeptide tri-links, and 122 were dipeptide bi-links. Overall, tri-linked cross-links contributed ~27% (45/167) of the total unique CSMs. Breaking down tripeptide and dipeptide tri-links into their respective constituent residue pairs, a combined total of 118 K-K pairs were identified across all CSMs, with 37 being contributed by both TSTO tri- and bi-links, whereas 50 and 31 were unique contributed by TSTO bi-links and tri-links, respectively.

Considering the spacer arm length of TSTO (~14 Å), lysine residues with C α -C α distance ≤ 35 Å were expected to be preferentially cross-linked. When mapped to the high-resolution crystal structure of BSA (PDB:4F5S) (Figure S4A), the overall mapped distance median was 21.1 Å with a satisfaction rate of cross-links under ≤ 35 Å of 90%. Taken together, these results demonstrate that TSTO is effective for protein cross-linking and the resulting cross-linked peptides exhibit unique MS² fragmentation patterns that are both predictable and reliable for unambiguous identification by LC MSⁿ analysis. ”

In addition, we have provided additional Supplemental Figures S1-S3 to show MS² and MS³ fragmentation of TSTO cross-linked synthetic peptide SR8 and BSA peptides, respectively.

Q2: *There is no word how specific the crosslinker is (other potential targets?) and potential side reactions. (Serine, threonine?) standard NHS ester.*

A2: Although we understand the reviewer's concerns, NHS ester chemistry and its potential side reactions in relation to cross-linking conditions have been well-established. While it has been suggested in the literature that NHS esters have the potential to target serine and threonine residues, we have observed minimal off-target serine and threonine cross-linking under our experiment conditions for all the NHS ester linkers that we have characterized over the years. Furthermore, a recent comparative study has shown that STY residues rarely react with NHS ester-based cross-linkers (Cao, *JPR*, 2023). To further confirm this, we have re-submitted database searches for four sets of SEC-HpHt MS acquisitions (24 samples) using TSTO modifications on lysine, serine, and threonine residues. Compared to the original searches performed on lysine residues alone, the KST search only yielded one additional CSM that contained a peptide with no internal lysine. All other cross-linked peptides potentially identified with an S/T residue also contained a nearby internal lysine. Based on these results, we feel that it is sensible to only present lysine cross-links—as reported by other XL-MS studies using NHS ester-based cross-linkers in the literature.

Q3. No control reactions reported that would show no cross links when lysines are modified or mutated.

A3: As stated in the response to Q2, the lysine targeting specificity of NHS esters has been well characterized and is not expected to change due to its incorporation within TSTO. As such, modified and mutated lysine residues would not be expected to be cross-linked. TSTO cross-linking has been characterized using a standard peptide and a model protein and the results have been included in the text. The validity of TSTO cross-links is supported by existing structures and the results corroborate well with previous XL-MS studies using other cross-linkers such as DSSO.

Q4. The 26S particles were affinity purified with Rpn1 but their quasi stability has not been addressed. The concern here is that many potential links are missing or even spurious when a complex and dynamic protein as proteasome is tested with crosslinkers.

A4. The reason we have selected the 26S proteasome for characterization of TSTO is that proteasome complexes have been broadly utilized not only by our lab but others to illustrate cross-linker applicability for XL-MS analysis of protein complexes (Tomko, *Mol Cell*, 2010; Kao, *MCP*, 2011; Kao, *MCP*, 2012; Leitner, *PNAS*, 2014; Tomko, *Cell*, 2015; Wang, *MCP*, 2017; Yu, *MCP*, 2019; Mendes, *Mol Syst Biol*, 2019; Yu, *Anal Chem*, 2022). In addition, our lab has developed multiple proteomics approaches to purify and study proteasome complexes, including affinity purification methods. Here, we would like to clarify that the 293 cells stably expressing HTBH-Rpn1 were specially made through CRISPR/Cas9-mediated genome editing. This ensures that HTBH-tagged Rpn1 protein is the only copy in the cell and that its expression is at the endogenous level (Lin, *J Bio Chem*, 2019), preventing artifacts caused by protein overexpression associated with conventional affinity purification. In addition, the 26S proteasome purified via HTBH-tagged Rpn1 has been well-characterized, and its integrity and function were confirmed by peptidase activity and mass spectrometric analysis. Therefore, we believe this cell line is well-suited for isolating the human 26S proteasome for characterizing TSTO.

To compare TSTO with bifunctional cross-linkers, we followed the XL-MS protocol that has been previously established for MS³-based analysis using DSSO, resulting in a TSTO cross-link identification FDR of 0.04%. Importantly, the observed cross-linking patterns were consistent with previously reported structures and XL-MS results, supporting their biological relevance. While it

is recognized that the proteasome is a dynamic and complex entity, our approach focuses on capturing interactions within its quasi-stable state under defined experimental conditions. Missing or spurious links were mitigated by low FDR and biological replicates, ensuring the reliability of our findings. While it would be interesting to illustrate quasi-dynamics of proteasomes in the future, such a study is beyond the scope of our current manuscript.

Q5. Trifunctional crosslinkers like TSTO shows obvious limitations arising from the necessity to simultaneously react with three residues placed at a proper distance. It has not been addressed how reaction time with TSTO affects the detected links and if protein dynamics could be deduced from the detected links (allosteric forms?).

A5: We appreciate the reviewer's input. While TSTO is designed to simultaneously react with three residues, we want to emphasize that it is not necessary for all three reactive groups to be utilized for cross-linking. Thus, TSTO cross-linking can result in three types of cross-link products: 1) trivalent cross-links in which all three NHS esters react with three lysines; 2) bivalent cross-links in which only two NHS esters of TSTO react and the remaining one is hydrolyzed; and 3) dead-end products (aka, monolinks) in which only one NHS ester of TSTO reacts and the other two are hydrolyzed.

As shown with standard peptide testing using Ac-SR8, all three types of cross-linked products were observed. We further tested labeling time using Ac-SR8, both in DMSO and in PBS buffer. In DMSO, 5 minutes was sufficient for labeling the majority (~80%) of the synthetic peptide, and ~30 minutes was sufficient for nearly complete labeling by TSTO. In PBS, 5 minute incubation of TSTO with the synthetic peptide yielded 50% labeling, likely due to the competing hydrolysis reaction in water; after 1 hour, the cross-linking efficiency was ~70%. Nevertheless, these results show that the NHS esters to be relatively active. As a result, TSTO (as well as other popular cross-linkers) can be used to study protein dynamics through stabilization of different conformational states. For dynamics that occur on a timescale similar to the cross-linking reactions, time-resolved XL-MS or combining with pulse-labeling strategies could potentially capture allosteric intermediates or detect conformational shifts. However, capturing short-lived conformational states with higher precision may require faster reactive chemistries, such as photo-reactive functional groups, to detect such dynamic rearrangements. We agree that how cross-linking mass spectrometry contributes to the analysis of protein dynamics is an interesting area that needs careful investigation in the future.

Q6. Although the Authors claim that the trivalent cross linker is better than tetra or di-valent but formal proof of that is absent.

A6: We apologize for the confusion and would like to clarify that the development of TSTO was aimed to complement existing bifunctional reagents by providing additional information to enhance our understanding of protein interactions and structures, not to replace them. This is due to the fact that no single reagent has the capability to generate a complete picture of human interactomes. We would like to emphasize that TSTO's trifunctional cross-linking capability can complement existing reagents as it allows the capture and identification of additional interactions that cannot be easily determined using existing bifunctional cross-linkers. This would expand the capability to define protein-protein interactions and heterogenous protein complexes. For example,

the detection of tri-peptide cross-links would allow us to distinguish between trimeric and dimeric PPIs:

Protein complex			Divalent cross-links				$\alpha\text{-}\alpha$	$\alpha\text{-}\beta, \beta\text{-}\beta$	$\alpha\text{-}\gamma, \beta\text{-}\gamma$
Trivalent cross-links				$\alpha;\alpha;\alpha$	$\alpha;\beta;\beta$	$\alpha;\beta;\gamma$

Figure 2. Diagram illustrating the interactions captured by trifunctional over bifunctional cross-linkers for characterizing trimeric protein complexes.

As illustrated in Figure 2, for a homotrimeric complex 3α , bivalent cross-linking would result in $\alpha\text{-}\alpha$ pairs. Despite multiple interactions that can be formed across neighboring α proteins, there lacks direct evidence supporting the presence of a homomeric 3α trimer. In comparison, a trivalent cross-link $\alpha;\alpha;\alpha$ would signify an interaction among three proteins, suggesting a trimer. The same analogy would apply to trimeric complexes involving a single homodimer ($\alpha\text{-}2\beta$) or heterotrimers ($\alpha\text{-}\beta\text{-}\gamma$).

To further determine the benefits of trivalent cross-linking, we have performed new integrative structural modeling to demonstrate that trivalent cross-links can improve the precision and accuracy of structural modeling of protein complexes compared to bivalent ones. This has been illustrated by structural modeling of proteasome subcomplexes based on both experimental cross-linking data, as well as a synthetic dataset to directly compare the model precision and accuracy obtained by trivalent versus bivalent cross-links. This new analysis has resulted in a new figure (Figure 4), which significantly strengthens our paper and has been included in the revision as follows:

“Integrative Structure Modeling using Synthetic and Experimental XL-MS Data

To determine the value of trivalent versus bivalent cross-links, we applied integrative structure modeling to compute the structure of the proteasome base subcomplex (subunits Rpt1–Rpt6 and Rpn2). First, we generated synthetic datasets for the proteasome base subcomplex,

replicating the experimental distance distribution of the proteasome TSTO dataset (Table S2) to systematically compare trifunctional and bifunctional cross-linkers. Subsets of 20, 30, 40, 60, and 120 cross-links were randomly sampled based on the distance probability distribution, with at least five replicates created for each subset size. For this analysis, a cross-linking site was defined as the set of residues bridged by a single cross-linker. To illustrate the uniqueness of trifunctional linkages, we have focused our analysis on the comparison of trivalent and bivalent cross-links for integrative modeling. For each trivalent cross-link, a corresponding bivalent cross-link utilizing two of the three cross-linked lysines was used to ensure that the synthetic data reflects the spatial restraints of each cross-linker, providing a robust foundation for structural modeling.

Ensembles of the proteasome subcomplex configurations that satisfy the input information (i.e., the model) were found by exhaustive Monte Carlo sampling guided by the scoring function, starting with random initial configurations of the rigid components. The models computed using trivalent cross-links were generally more accurate and precise than the ensembles computed using bivalent ones. The accuracy is defined as the average C α root-mean-square deviation (RMSD) between the cryo-EM structure (PDB ID: 5GJR) and each of the structures in the ensemble, while the precision is defined as the average RMSD between all solutions in the ensemble. Increasing the number of both trivalent and bivalent cross-links used for modeling improved model accuracy and precision, with diminishing returns observed as the number of cross-links increased from 40 to 120. For datasets containing bivalent cross-links, the average model accuracy plateaued at ~9.8 Å, while for trivalent cross-links the plateau occurred at ~8.7 Å. The trivalent cross-linking data consistently produced better model accuracies, with average RMSDs of 19.7, 14.1, 9.7, 8.8, and 8.7 Å for datasets with 20, 30, 40, 60, and 120 cross-links, respectively. In contrast, the bivalent cross-linking data resulted in higher average RMSDs of 25.3, 19.0, 13.3, 9.7, and 9.8 Å for the same subset sizes. A similar trend was observed for the cluster precisions (Figure 4). Furthermore, we computed the structure of the base subcomplex of the 26S human proteasome using integrative modeling with experimentally derived DSSO and TSTO cross-linking datasets. The DSSO dataset included 83 cross-links within the base subcomplex. The TSTO dataset comprised 18 trivalent and 143 bivalent cross-links; from the TSTO data, 65 bivalent cross-links were randomly selected (five replicates) to ensure that the total number of cross-linking sites was consistent between the DSSO and TSTO datasets. Models computed using the TSTO dataset were considerably more accurate than models computed with the DSSO dataset (15.3 Å vs 33.2 Å). These results highlight the ability of the TSTO cross-linker to provide richer structural information, such as the spatial positioning of three protein regions, which is critical for generating accurate structural models. ”

To the best of our knowledge, the only published multivalent cross-linker is the tetrafunctional reagent called Bisby (Mohr et al., *J Proteome Res*, 2023). While it can capture four interactions to provide more information than trifunctional cross-linker TSTO, their identification remains challenging due to the nature of the cleavable bonds incorporated into the linker, which are peptide bonds. In comparison, the cleavable bonds in TSTO are weaker than peptide bonds and can be preferentially cleaved during low energy collision-induced dissociation (CID). Importantly, the unique cleavability of TSTO allows simultaneous release of the three cross-linked ends during CID, minimizing the number of fragments for MS³ sequencing and thus simplifying data acquisition and analysis for accurate cross-link identification. In contrast, the four cleavable bonds incorporated in the Bisby design cannot be cleaved simultaneously, resulting in a mixed population

of fragments including ones with multiple peptide constituents, which significantly complicates MS analysis for cross-link identification. While the feasibility of Bisby has been demonstrated for XL-MS analysis, its application for proteome-level studies has not been reported. Therefore, TSTO represents the first and robust MS-cleavable trifunctional cross-linker that is suited for capturing multimeric interactions at the proteome scale.

Q7. It is surprising that in an XL proteome of HEK293 cells 1512 proteins were found but only 1242 interactions were revealed. Is it an indication of the limitation of TSTO reactivity?

A7: We thank the reviewer for their comment. Based on immunoblot analysis of cross-linked HEK293 cells, TSTO appears to be highly reactive as the concentration of TSTO (500 μ M~1mM) required for effective formation of cross-linked protein oligomers is less than that of other popular bifunctional cross-linkers (2~10 mM) used for in-cell cross-linking. Therefore, we suspect that the low number of interactions revealed relative to the number of proteins identified is most likely due to the heterogenous populations of cross-links resulting from TSTO cross-linking. Based on our data, ~35% of inter-protein PPIs were commonly identified between bivalent and trivalent cross-links, with 62% of all inter-protein PPIs identified in trivalent cross-links being covered by bivalent ones. This is most likely attributed to the reduced number of unique interactions identified in the dataset, but each interaction was identified with higher confidence as they were supported by an increased number of cross-links. This aspect has been included in the discussion as follows:

“While TSTO is highly effective for in-cell cross-linking at lower concentrations than bifunctional cross-linkers (Wheat, *PNAS*, 2021; Chavez, *Nat Protoc*, 2019; Jiang, *Angew Chemie*, 2022), the heterogeneity of TSTO cross-links appears to impact the number of unique PPIs obtained compared to bivalent cross-linkers at the proteome-wide scale. Based on our data, 62% of all inter-protein PPIs identified in trivalent cross-links were covered by bivalent ones. This is most likely attributed to the reduced number of unique interactions identified in the current dataset. It is noted that each interaction was identified with higher confidence as they were supported by an increased number of cross-links. In addition, as the integrative structural modeling has showed, trivalent cross-links are advantageous in providing additional spatial information to facilitate structure modeling with increased precision and accuracy.”

Q8. The results presented in Figure S4 need more complete association with the stipulated cell permeability. The selection of 1mM TSTO also needs better explanation. The immunoblot shows monomer of casein and after the crosslinking only a band of very large aggregates (?) but no other oligomers.

A8. We apologize for the confusion. To evaluate in-cell cross-linking efficiency, we have followed our previously published protocol by probing the formation of cross-linked protein complexes containing a selected target using western blot (Wheat, *PNAS*, 2021). Here, the HEK293 cells used for in-cell TSTO cross-linking stably express HB-tagged CSN2 protein, a subunit of the COP9 signalosome complex. To evaluate TSTO in-cell cross-linking efficiency, we performed immunoblot analysis to probe the formation of CSN2-containing cross-linked products, which is represented by the high molecular weight protein bands. The higher relative abundance of CSN2-containing oligomer to its monomer suggests higher cross-linking efficiency. Based on the western blot, 1 mM TSTO appears to provide sufficient yield of cross-linked proteins. We have clarified this in the figure legend for Figure S4 and described it accordingly in the text:

“To evaluate TSTO in-cell cross-linking efficiency, we performed immunoblot analysis to probe the formation of CSN2-containing cross-linked products, represented by high molecular weight protein bands. Based on the increasing relative abundance of CSN2-containing oligomer to its monomer with increasing cross-linker concentration, TSTO was determined to be membrane-permeable and suited for in-cell cross-linking (Figure S4).”

Q9. I would recommend inclusion of the major findings in the abstract. Currently, we are only treated to an introduction of TSTO with a very general advertisement of its power.

A9. We appreciate the reviewer’s suggestion for improving the abstract. We have included a description of the major findings in the abstract:

“Cross-linking mass spectrometry (XL-MS) is a powerful technology for mapping protein-protein interactions (PPIs) at the systems-level. By covalently connecting pairs of proximal residues, cross-linking reagents provide distance restraints to infer protein conformations and interaction interfaces. While bivalent cross-links have been remarkably informative, multivalent cross-links can offer enhanced spatial resolution to facilitate the characterization of heterogeneous protein complexes. However, their identification remains extremely challenging due to fragmentation complexity and vast expansion of database search space. Here, we present a novel trioxane-based MS-cleavable homotrifunctional cross-linker TSTO, which can target three proximal lysine residues simultaneously. The unique MS-cleavability of TSTO enables concurrent release of cross-linked peptide constituents during low-energy collision induced dissociation, simplifying LC-MSⁿ analysis for their unambiguous identification. Importantly, we have demonstrated that the TSTO-based XL-MS platform is effective for mapping multimeric interactions of protein complexes. Moreover, TSTO has been shown to be well-suited for in-cell and in-tissue XL-MS studies for elucidating cellular networks, illustrating its versatility for complex biological systems. Furthermore, the trimeric interactions captured by TSTO have uncovered new structural details that cannot be easily revealed by bifunctional reagents, which augmented structural modeling with improved accuracy and precision. This development not only advances XL-MS technologies for global PPI profiling but also offers a new direction for creating multifunctional MS-cleavable cross-linkers to further push structural systems biology forward in the future.”

Q10. Very minor: in Fig 2A labels of subunit shifted.

A10. Thank you and we have adjusted the legend positioning accordingly with their corresponding subunit designations.

Q11. Very minor: the mw ladder band representing 50kDa is out of the position – the distance between 50 and 60 kDa is much larger than between 40 and 50 kDa. Is this a gradient gel?

A11. The gels used here for SDS-PAGE are 10% acrylamide gels. We typically see a separation between 50 and 60 kDa when using PageRuler™ Unstained Protein Ladder (Thermo) that is slightly larger than expected.

Reviewer #4

Q1: *The trioxane derivative MS-cleavable cross linker allows an MS3 clear identification of targeted fragments after digestion. The authors define properly this very interesting tool, firstly by detailing the chemical synthesis of the TSTO XL-linker. Then, they highlight the tagging effect through the detection of specific fragment ions signature by MS2 generation of α , β , γ modified peptides coupled with their * ions corresponding to the loss of H₂O. This “featured” signature may potentially not be sufficiently unambiguous and should be more discussed and illustrated.*

A1: We thank the reviewer for pointing this out. While the signature of TSTO fragmentation itself is not a prerequisite for identification (as shown by results using non-targeted DDA analysis), the frequency of the water-loss event can be utilized for targeted ion selection to improve cross-linked peptide analysis. As the reviewer points out, H₂O-loss is not a unique feature of TSTO and can be observed during normal peptide backbone cleavage. However, water loss peaks in MS² analysis resulting from peptide backbone cleavage are limited due to the unique characteristics of TSTO cross-links. As TSTO is preferentially cleaved prior to the breakage of the peptide backbone, we have observed that the dominant pairs with $\Delta m = 18.01$ Da in the MS² spectra of TSTO cross-links correspond to separated AR- and AR*-modified cross-linked peptides.

When directly comparing between the two types (top intensity-based and targeted mass-difference) of acquisition methods for MS³, ion selection using H₂O-loss yielded a modest increase in cross-link identification due to improved peptide backbone fragmentation. This is due to the fact that both AR- and AR*-modified peptides are selected for MS³ analysis in top intensity-based acquisitions. However, the dominant ion resulting from fragmentation of AR-modified peptides in MS³ spectra is the AR* moiety, alongside minimal peptide backbone fragmentation. In comparison, fragmentation of AR*-modified peptides results in significantly richer MS³ spectra for improved peptide identification. To illustrate this, we have included characterization of TSTO in a synthetic peptide, showing the difference in MS³ fragmentation between the two TSTO remnant moieties. (Figure S1). Using the mass-difference targeted approach, AR*-modified peptides can be preferentially selected to ones carrying AR, resulting in improved cross-link identification.

Q2: *The authors present a very interesting proof of concept using the model of 26S Proteasomes interactomic study with the identification of 808 unique CSMs and 41% of tri-linked species. However, Authors should more particularly describe the gain of information compared to bifunctional XL-MS platforms.*

A2: We thank the reviewer for their suggestions. To expand on the gain of information acquired by TSTO cross-linking, we have compared cross-link identifications against a previous dataset acquired using DSSO cross-linking on 26S proteasomes purified using the same tagged 19S subunit. For both unique K-K cross-links and PPI interactions, the majority of the DSSO dataset was covered by TSTO. Specifically, 70% of all inter-subunit and 94% of all intra-subunit cross-links identified by DSSO were covered by TSTO. In terms of PPIs, 94% of all inter-subunit and 100% of all intra-subunit PPIs identified by DSSO were also identified by TSTO.

Moreover, to illustrate the benefits of TSTO in structural characterization, we have performed new integrative structural modeling experiments to demonstrate the improved precision and accuracy

offered by trivalent cross-links versus bivalent cross-links using both synthetic and experimental cross-linking data. The results of this new analysis have been included as a new figure (Figure 4) and are detailed in the response to question #6 from Reviewer #2. In summary, our new experiments and analyses have further demonstrated the unique benefits offered by TSTO in yielding multimeric interactions to provide additional spatial resolution to facilitate structural modeling and characterization of heterogenous protein complexes.

Q3: Although the results are relevant and clearly reported, the present strategy may not be sensitive enough to be used in an endogenous context, because of an obvious loss of sensitivity due to the multiplicity and diversity of signals. The authors don't illustrate the possibility of obtaining monomeric species, which represents a fourth type of "cross-link".

A3: We thank the reviewer for their input. Similar to other NHS ester-based cross-linking reagents, obtaining monomeric cross-linked species is a given due to the competing hydrolysis reaction in aqueous solution. As described in the previous comments, we have provided additional supplemental figures and text describing the characterization of TSTO, which includes formation and cleavage of monomeric 'dead-end' cross-linked species. However, as these types of cross-links are generally less informative for PPI mapping, we did not elaborate on their identifications here. Furthermore, many monomeric species are lost during the initial peptide SEC step, which was performed to remove 'dead-ends/monolinks' and 'intra-links/loop-links' and improve the identification of di- and trivalent inter-linked species. We agree that these linear cross-linked peptides can be informative for identifying solvent-exposed regions of the proteins and aid in their structural elucidation, which can be further investigated in future studies. Here, we focus on the elucidation of protein-protein interactions that are derived from bivalent and trivalent cross-link products. To clarify, the possible cross-linked products resulting from TSTO have been described in the text:

“While monomeric cross-linked species such as 'dead-end' and intra-links can occur as a result of NHS ester hydrolysis of two groups and cross-linking of internal lysines within the same peptide, the three types of TSTO inter-protein cross-links are the most structurally informative products for PPI mapping and can be identified using the same LC-MSⁿ workflow that has been previously established for our sulfoxide-containing MS-cleavable cross-linkers.”

Q4: Finally, exploration of the multiplex potential of this tool could be of primary importance. The discussion section should be expanded on this point, and the results should be demonstrated more thoroughly, emphasizing the potential of this technique in real in vivo conditions, as HEK293 cell culture may not be representative of an in vivo model.

A4: We thank the reviewer for their suggestion. Regarding the multiplex potential of TSTO, we believe that the compatibility of using isobaric labeling and/or metabolic labeling methods (i.e. SILAC) would be similar to other NHS ester-based bifunctional cross-linkers such as DSSO. Incorporating isotopic labels in the TSTO design would offer additional capability for quantitative XL-MS analysis. We have added additional text in the discussion to describe the potential for multiplexing TSTO experiments:

“Beyond improving structural characterization and expanding the scope of detectable interactions, the TSTO-based XL-MS platform can be coupled with isotope-based quantitative strategies to

enable the interrogation of how multimeric protein interactions fluctuate in response to biological stimuli, disease states, or drug treatments, providing a deeper functional understanding of cellular organization. The isotopic labels can be introduced into TSTO cross-links by SILAC labeling of lysines(Yu, *MCP*, 2019), chemical labeling of cross-linked peptides with isobaric reagents (e.g. TMT)(Yu, *Anal Chem*, 2016), or coding isotopic labels in the linker design.”

We understand the reviewer’s concern on whether HEK293 cells would be a good system as an *in vivo* model. However, we would like to point out that mammalian cell lines have been commonly used in XL-MS studies to demonstrate cross-linkers’ capability for *in vivo* cross-linking analyses (Wheat, *PNAS*, 2021; Jiang, *Angew Chemie*, 2022; Gao, *Anal Chem*, 2023). Nonetheless, to further demonstrate the potential of TSTO for *in vivo* studies, we have performed in-tissue cross-linking of mouse heart. The results are described in a new section of the text as follows:

“In Vivo TSTO Cross-linking of mouse heart tissue

To illustrate the feasibility of TSTO in another biological context, we performed in-tissue cross-linking to identify cross-linked peptides using the same workflow for in-cell XL-MS as established above. Here, TSTO cross-linking of intact mouse hearts resulted in the identification of a total of 4,770 CSMs using LC MSⁿ. Among these, 170 unique trivalent and 921 unique bivalent cross-link CSMs were identified, providing a snapshot of protein interactions within the heart. The most abundant XL-PPIs involve proteins that are major structural and functional components of cardiac tissue (Figure S12A), including actin, myosin, tropomyosin, and ATP synthase subunits. This observation correlates with the known role of these proteins in heart muscle function, particularly in the sarcomere—the contractile unit of muscle fibers.

To further explore protein interactions within the cardiac muscle, we generated an XL-PPI map, focusing on the interactions of cardiac actins with myosins, tropomyosins, and troponins, all of which are key regulators of muscle contraction (Figure S12B). Myosin, a motor protein, binds actin and its function is regulated by tropomyosin and troponin. Tropomyosin, a coiled-coil protein, runs along actin filaments and controls myosin access to actin-binding sites, while the troponin complex (composed of troponin C, T, and I) binds actin and tropomyosin to modulate contraction by controlling myosin-binding site exposure. Consistent with these molecular functions, our cross-linking data revealed interactions among these proteins, with the most direct connections observed between myosin and actin, while tropomyosin and troponins were cross-linked with both actin and each other (Figure S12B). Notably, trivalent cross-links were identified not only within myosins and tropomyosins but also between troponin and actin, as well as between actin and tropomyosin (Table S3). Additionally, we found that tropomyosin was cross-linked to an ATP synthase subunit (Table S3). While this interaction is not typically associated with sarcomere function, it could suggest a potential regulatory link between contractile and energy-producing processes, possibly reflecting coordinated control of ATP synthesis and muscle contraction in cardiac cells. Further Gene Ontology (GO) enrichment analysis of heart-specific proteins captured by TSTO cross-linking revealed a significant representation of pathways associated with cellular energy metabolism, particularly those involved in ATP synthesis and mitochondrial function, which are critical for maintaining cardiac contractility and overall heart function (Figure S12C-E). Taken together, our results have shown the feasibility of TSTO for in-tissue cross-linking, expanding its applicability for different biological samples.”

To address the reviewer's comments, we have also added an additional paragraph in the discussion describing tissue cross-linking and potential strategies to improve the protein identification depth obtained through TSTO cross-linking:

“It is noted that the overall coverage of in-tissue XL-PPIs was less extensive compared to in-cell XL-PPIs revealed by TSTO. This is more likely due to the higher complexity and lower recovery of tissue proteins and cross-linked peptides. To enhance protein coverage and increase cross-link identification from proteome-wide samples, fractionation of cross-linked protein complexes can be coupled with peptide separation to improve analysis sensitivity and dynamic range. Although cross-link enrichment through incorporation of affinity tags such as biotin, azide, or phosphonic acid have been successful with previous bifunctional cross-linkers, this approach is likely impractical for TSTO due to the central positioning of its trioxane functional group. However, recent developments in polyclonal antibodies that can recognize MS-cleavable cross-linkers (Singh, *Anal Chem*, 2021) suggest that similar affinity purification strategies could be adapted to enrich TSTO cross-links and increase their MS detectability. Moreover, while MS³-based analysis allows accurate cross-link identification, it is less sensitive than MS² analysis. Thus, MS²-based workflows could be explored to further increase the sensitivity of TSTO cross-link identification. This will require the development of new software to enable the identification of three cross-linked peptides from a single MS² spectrum. Due to the unique MS-cleavability of TSTO, we believe MS²-based identification of TSTO cross-links would be highly feasible compared to the identification of tri-links formed by conventional bifunctional cross-linkers. Given its unique ability to capture trivalent cross-links and its robustness in cross-link identification, TSTO represents a promising cross-linker that can provide additional spatial restraints to significantly enhance our understanding of protein modules and their organization in complex biological systems.”